# DNA methylation-free *Arabidopsis* reveals crucial roles of DNA methylation in regulating gene expression and development

Li He [1], Huan Huang[1], Mariem Bradai[1], Cheng Zhao[1,2], Yin You[1,3], Jun Ma[1], Lun Zhao[1], Rosa Lozano-Durán[1,4] & Jian-Kang Zhu [1,5 ✉]

A contribution of DNA methylation to defense against invading nucleic acids and maintenance of genome integrity is uncontested; however, our understanding of the extent of involvement of this epigenetic mark in genome-wide gene regulation and plant developmental control is incomplete. Here, we knock out all five known DNA methyltransferases in *Arabidopsis*, generating DNA methylation-free plants. This quintuple mutant exhibits a suite of developmental defects, unequivocally demonstrating that DNA methylation is essential for multiple aspects of plant development. We show that CG methylation and non-CG methylation are required for a plethora of biological processes, including pavement cell shape, endoreduplication, cell death, flowering, trichome morphology, vasculature and meristem development, and root cell fate determination. Moreover, we find that DNA methylation has a strong dose-dependent effect on gene expression and repression of transposable elements. Taken together, our results demonstrate that DNA methylation is dispensable for *Arabidopsis* survival but essential for the proper regulation of multiple biological processes.

[1] Shanghai Center for Plant Stress Biology, Center for Excellence in Molecular Plant Sciences, Chinese Academy of Sciences, 201602 Shanghai, China. [2] Department of Clinical Science Intervention and Technology, Karolinska Institutet, 14186 Stockholm, Sweden. [3] University of the Chinese Academy of Sciences, 100049 Beijing, China. [4] Department of Plant Biochemistry, Centre for Plant Molecular Biology (ZMBP), Eberhard Karls University, D-72076 Tübingen, Germany. [5] Present address: Institute of Advanced Biotechnology and School of Life Sciences, Southern University of Science and Technology, 518055 Shenzhen, China. ✉email: zhujk@sustech.edu.cn

D NA methylation at the C-5 position of cytosine residues is a conserved epigenetic modification involved in transposon silencing, genome stability, and regulation of gene expression in many eukaryotes[1–13]. In plants, DNA methylation occurs in three sequence contexts: CG, CHG, and CHH, where H is any nucleotide except G. Plants have evolved different DNA methyltransferases for the establishment and maintenance of DNA methylation depending on the cytosine sequence contexts[13,14]. CG methylation is maintained by METHYL-TRANSFERASE 1 (MET1), an orthologue of the mammalian DNA methyltransferase 1 (DNMT1), which recognizes hemimethylated CG dinucleotides following DNA replication and methylates the unmodified cytosine in te daughter strand[15,16]. Maintenance of CHG methylation is catalyzed by CHROMO-METHYLASE 3 (CMT3) and, to a lesser extent, by CMT2 and is tightly linked to dimethylation of lysine 9 on histone 3 (H3K9me2)[17–19]. The asymmetric CHH methylation is maintained by two different types of DNA methyltransferases, CMT2 and DOMAINS REARRANGED METHYLTRANSFERASE 1/2 (DRM1/2, homologs of DNMT3), depending on the genomic regions: CHH methylation in heterochromatin is maintained by CMT2, whereas CHH methylation in euchromatin or at the edge of long transposable elements (TEs) is mediated by DRM1/2 through the RNA-directed DNA methylation (RdDM) pathway[20–24]. RdDM is also important for de novo DNA methylation in all three sequence contexts[1,2,21,23,25–28]. Even though the roles of these five DNA methyltransferases have been extensively studied, whether they are sufficient to maintain DNA methylation throughout the entire genome or, on the contrary, the contribution of other yet-to-be-described DNA methyltransferases is required, is unresolved.

Heavy DNA methylation occurs in heterochromatic regions, which are enriched with TEs and other repetitive DNA sequences[29–31]. Extensive hypomethylation in heterochromatic regions causes genome-wide TE transcriptional activation and induces TE transposition after continuous selfing[20,32–39]. Thus, DNA methylation plays an important role in repressing invading nucleic acids (transposable elements, viruses, and transgenes). In contrast to TEs, which are methylated in their body region in all three sequence contexts, gene-associated DNA methylation can occur in the promoter region or in introns, or within the transcribed gene body only in the CG context. DNA methylation in different positions within genic sequences seems to play different functions: (1) in promoters, DNA methylation usually inhibits gene transcription, although the opposite effect has occasionally been observed[29,40,41]; (2) genes with DNA methylation in their body in the CG context are generally constitutively expressed[29,42,43]; and (3) DNA methylation in the introns of several genes has been shown to promote polyadenylation of their full-length transcripts[44–49], but whether intronic DNA methylation has a general effect on transcript processing is still uncertain. In order to fully assess the genome-wide contribution of DNA methylation to gene expression, the generation of stable DNA methylation-free plant materials is required. Although mutant Arabidopsis plants displaying a nearly complete loss of either CG or non-CG methylation are available[19,29], mutants with full erasure of DNA methylation have not been reported.

DNA methylation is critical for development in animals: e.g., null mutation in DNMT1 or DNMT3a/3b in mice leads to embryonic lethality[13]. However, loss of function of MET1, DRM1/2, CMT2, or CMT3 does not result in obvious developmental abnormalities in Arabidopsis, with the exception of the late flowering phenotype displayed by met1[15,50]. The drm1 drm2 cmt3 cmt2 quadruple mutant, which loses almost all non-CG methylation, has curled leaves and a slightly reduced rosette size[19,51,52]. Simultaneous mutations in both met1 and some of the non-CG methyltransferases produces serious developmental defects, including extremely slow growth and late flowering, reduced plant size, and sterility[53,54]. Nevertheless, the lack of a met1 drm1 drm2 cmt3 cmt2 quintuple mutant has precluded conclusive studies regarding the contribution of DNA methylation to the regulation of different aspects of plant development.

Here, we generate the elusive Arabidopsis quintuple mutant in which the genes encoding all known functional DNA methyltransferases (MET1, DRM1, DRM2, CMT3, and CMT2) have been mutated (hereafter referred to as mddcc). mddcc plants exhibit extreme growth retardation and small size, display a suite of severe developmental defects, and do not undergo floral transition. We report the genome-wide analysis of DNA methylation, RNA expression, and TE movement in this mutant, demonstrating that: (i) DNA methylation in all contexts is completely eliminated in mddcc; (ii) DNA methylation exhibits a strong dose effect in terms of the regulation of gene expression and TE silencing; and (iii) CG and non-CG methylation act redundantly in the control of multiple biological processes, including flowering, trichome morphology, vasculature and meristem development, and root cell differentiation, likely through affecting the expression of genes encoding central regulators of these processes. Our findings extend prior knowledge on the importance of DNA methylation for global gene expression and development in plants and establish a framework for the future investigation of the contribution of this epigenetic mark to specific biological processes.

## Results

**MET1, DRM1, DRM2, CMT3, and CMT2 maintain the entire DNA methylation of the Arabidopsis genome.** To determine whether the five known DNA methyltransferases maintain the entire DNA methylation of Arabidopsis plants, we combined the power of CRISPR/Cas9-mediated genome editing with traditional genetics to knock out the corresponding protein-coding genes[55]. First, we crossed cmt2 to the drm1 drm2 cmt3 (ddc) triple mutant and isolated the drm1 drm2 cmt3 cmt2 (ddcc) quadruple mutant; then, we used a CRISPR/Cas9 system with two sgRNAs targeting the first exon of MET1 to generate a met1 mutation in the ddcc background. Cas9-free met1/+ drm1 drm2 cmt3 cmt2 seeds were obtained in the T2 generation, and met1 drm1 drm2 cmt3 cmt2 (mddcc) individuals were isolated from the progeny of these plants. We named the met1 allele generated in the quadruple mutant background met1-8 (Fig. 1a and Supplementary Fig. 1a). In parallel, the same CRISPR/Cas9 vector was used to produce a met1 single mutant; the allele generated in the WT background was named met1-9 (Fig. 1a and Supplementary Fig. 1a). met1-8 results in a frameshift mutation that creates a premature stop codon in the first exon of MET1 (Fig. 1a and Supplementary Fig. 1a). In met1-9, a 531 bp fragment was deleted from the MET1 gene, causing a deletion of 177 aa in the N-terminal part of MET1 (Fig. 1a).

To examine the DNA methylation levels in ddcc, met1-9, and mddcc mutants, we carried out whole-genome bisulfite sequencing (Supplementary Table 1). As expected, almost all CG and non-CG methylation is eliminated in met1-9 and ddcc mutants, respectively (Fig. 1b, c and Supplementary Fig. 1b–d and Fig. 2a, b). A more detailed observation of DNA methylation levels across chromosomes revealed that a low level of CG and non-CG methylation remains at pericentromeric regions in met1-9 and ddcc relative to mddcc, respectively, indicating that MET1 and DRM1/DRM2/CMT2/CMT3 can contribute to some extent to maintaining non-CG and CG methylation, respectively (Fig. 1b and Supplementary Fig. 2a). To investigate whether genomic DNA methylation is completely erased in mddcc, we compared

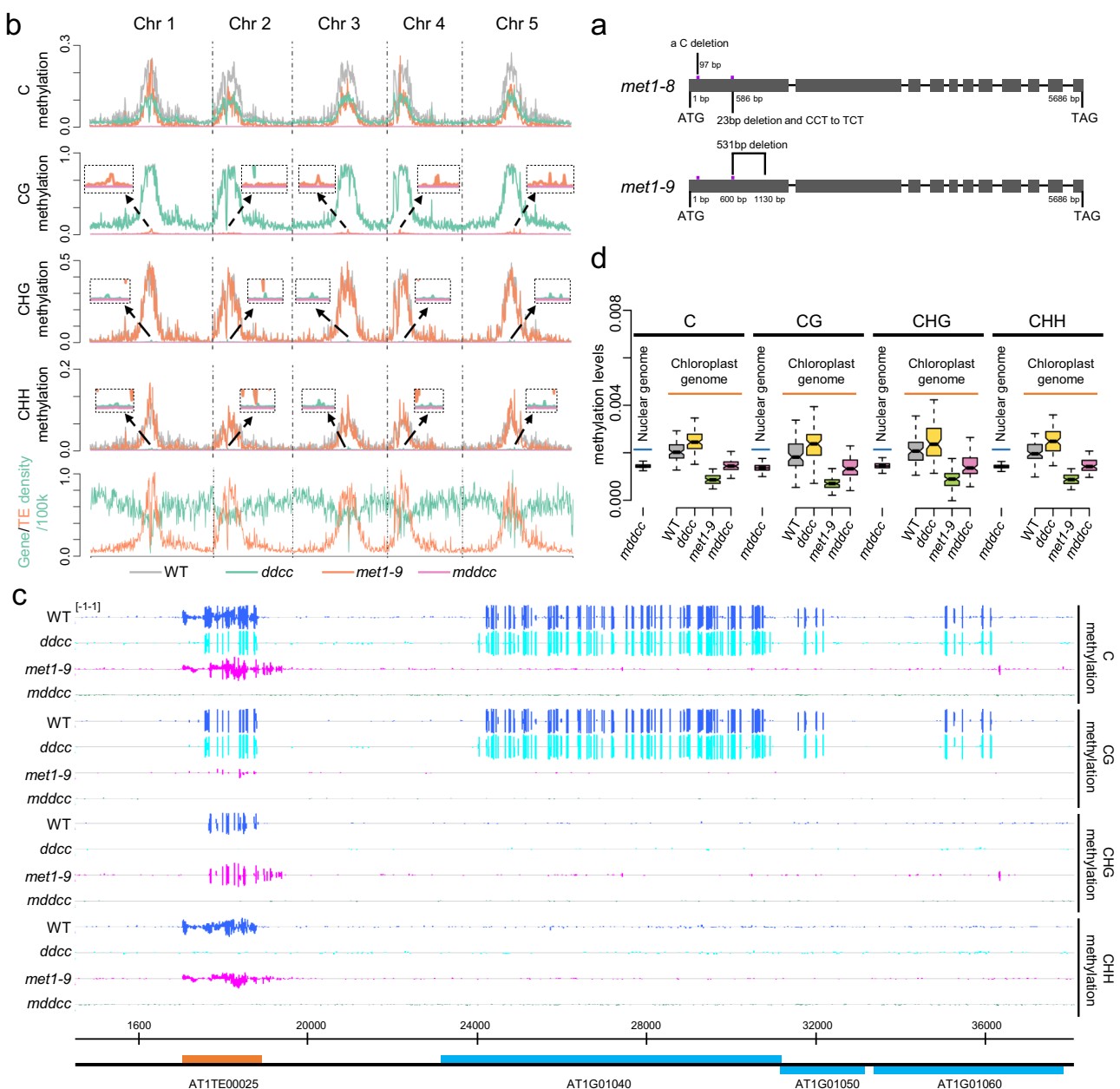

**Fig. 1 DNA methylation is eliminated in the *mddcc* mutant. a** Diagrams of the *MET1* gene showing the mutation sites in *met1-8* (upper) and *met1-9* (lower) mutants. Purple bars indicate the position of the sgRNAs used. **b** Genome-wide distribution of DNA methylation in all three sequence contexts in the indicated mutants. Wireframes are zoom-in of the indicated regions. Biological replicates were combined as one sample. Independent biological replicates are shown in Supplementary Fig. 2a. It should be noted that the lines of the WT overlap with those of *ddcc* in the mCG row. **c** Screenshot of DNA methylation levels over one representative locus in the indicated mutants. Orange and blue bars indicate TEs and genes, respectively. TEs and genes oriented 5′–3′ and 3′–5′ are shown above and below the line, respectively. Biological replicates were combined as one sample. Independent biological replicates are shown in Supplementary Fig. 2b. **d** Comparison of the DNA methylation levels between nuclear and chloroplast genomes in the indicated genotypes. Biological replicates were combined as one sample. Independent biological replicates are shown in Supplementary Fig. 2c. The horizontal line within the box represents the median; the whiskers represent minimum and maximum values; and the lower and upper boundaries of the box represent the 25th and 75th percentiles, respectively.

the DNA methylation levels of the nuclear genome with those of the methylation-free chloroplast genome. Noticeably, the DNA methylation of the nuclear genome in *mddcc* was as low as that of the chloroplast genome in any mutant background in all three sequence contexts (Fig. 1d and Supplementary Fig. 2c), indicating that MET1, DRM1, DRM2, CMT3, and CMT2 are responsible for the maintenance of the entire DNA methylation in the *Arabidopsis* genome, and implying that additional yet-to-be-described DNA methyltransferases do not contribute genome-

wide to this epigenetic modification (Fig. 1d and Supplementary Fig. 2c). Thus, the *mddcc* mutant is a DNA methylation-free *Arabidopsis* genotype.

**DNA methylation regulates gene expression in a dose-dependent manner.** We performed strand-specific RNA sequencing (RNA-seq) in WT, *ddcc*, *met1-9*, and *mddcc* (Supplementary Fig. 3a and Supplementary Table 2). By analyzing differentially expressed genes relative to WT (DEGs; fold change

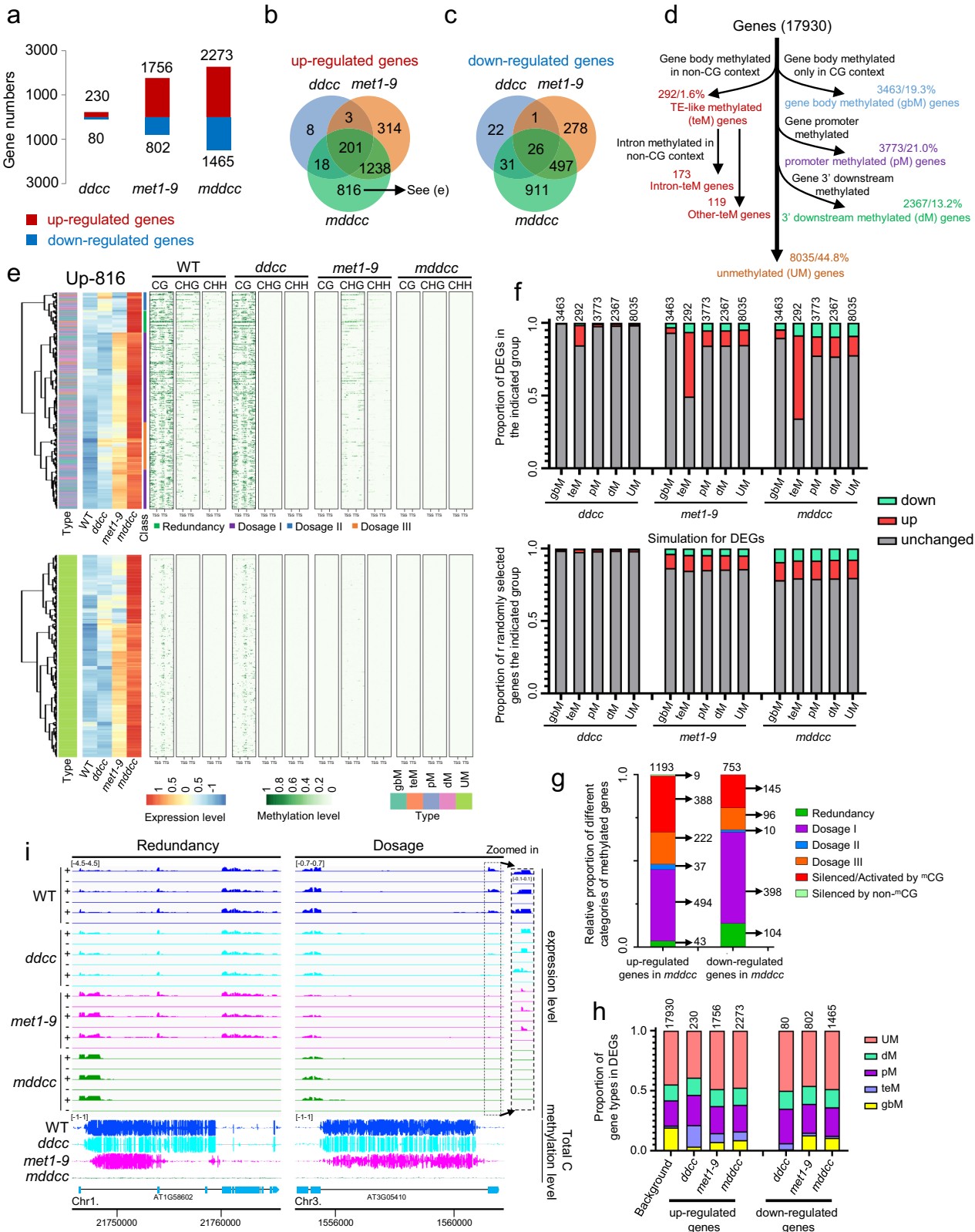

>2, p-adjusted < 0.01), we found that, in contrast to *met1-9*, which shows thousands of DEGs, there are just ~300 DEGs in *ddcc* (Fig. 2a), suggesting that the impact of CG methylation on gene transcription directly and indirectly is much stronger than that of non-CG methylation. Furthermore, the *mddcc* mutant shows a substantially higher number of DEGs than *met1-9* (Fig. 2a), supporting a partial functional redundancy between CG and non-

CG methylation in the regulation of gene expression. The subsets of DEGs in *ddcc*, *met1-9*, and *mddcc* highly overlap with one another (Fig. 2b, c). To confirm our identified DEGs, we randomly selected 25 up-regulated and 25 down-regulated genes in the *mddcc* mutant for validation by qPCR; the results obtained are consistent with the RNA-seq data, as shown in Supplementary Fig. 4. Gene ontology (GO) analysis showed that defense- and

**Fig. 2 DNA methylation regulates genes expression in a dose-dependent manner. a** Numbers of differentially expressed genes (DEGs) in the indicated mutants relative to WT. **b**, **c** Venn diagrams showing the overlap among up-regulated genes (**b**) or down-regulated genes (**c**) in *ddcc*, *met1-9*, and *mddcc* mutants. **d** Flow diagram indicating the criteria for gene classification according to the DNA methylation status in the WT background. Genes with no detectable expression (FPKM < 0.5) in any genotype were excluded. FPKM: fragments per kilobase of exon model per million mapped fragments. **e** Heat maps showing the expression and DNA methylation patterns of the indicated groups of up-regulated in DNA methyltransferase-deficient mutants. Based on their expression pattern, DEGs methylated in the WT were classified in the following categories: Redundancy (expression increased only in *mddcc*); Dosage I (expression mildly increased upon loss of CG methylation (in *met1-9*), and further increased upon loss of non-CG methylation (in *mddcc*)); Dosage II (expression mildly increased upon loss of non-CG methylation (in *ddcc*), and further increased upon loss of CG methylation (in *mddcc*)); and Dosage III (expression weakly increased upon loss of either CG or non-CG methylation (in both *met1-9* and *ddcc*) and further increased upon complete loss of DNA methylation (in *mddcc*)). **f** Proportion of DEGs in the gbM, teM, pM, dM, or UM categories in the indicated genotypes. Down: down-regulated; up: up-regulated. Numbers over the bars indicate the total number of genes in each category. Bottom, we randomly selected the same numbers of genes to perform the overlap analysis. **g** Relative proportion of different categories of methylated genes. The number of genes in each category is indicated. **h** Proportion of gene types in the indicated DEG subsets. Numbers above the bars indicate the total number of DEGs. **i** Snapshots of expression and DNA methylation levels over two Intron-teM genes in the indicated mutants. Structure of genes are shown in the bottom. + and − indicate forward and reverse strands in the genome, respectively. Source data are provided as a Source Data file.

response to stimuli-related GO terms are enriched in the subset of up-regulated genes in *ddcc*, *met1-9*, and *mddcc*, whereas metabolic process-related GO terms are enriched in the subset of down-regulated genes in *met1-9* and *mddcc* (Supplementary Fig. 5a, b). Interestingly, cell division-related GO terms are specifically enriched in the *mddcc*-unique down-regulated genes (Supplementary Fig. 5b).

To perform a more detailed analysis of the function of DNA methylation in the regulation of gene expression, we first classified the 17,930 genes with detectable expression (genes with no detectable expression in any genotype were excluded) into five main categories based on their methylation level in the WT background: 3463 (19.3%) genes with gene body methylation only in the CG context (gbM genes), 292 (1.6%) genes with gene body methylation in non-CG contexts (teM genes, TE-like methylated genes), 3773 (21.0%) genes with methylation in their promoters (pM genes), 2367 (13.2%) genes with methylation in their 3′ downstream regions (dM genes), and 8035 (44.8%) unmethylated genes (UM genes) (Fig. 2d). The proportion of DEGs among gbM genes is low (Fig. 2f). A significant proportion of DEGs was found among dM genes, suggesting that DNA methylation in the 3′ downstream of genes also plays a role in regulating gene expression (Fig. 2f). The proportion of up-regulated genes in the teM subset is significantly higher than that of down-regulated genes, indicating that non-CG gene body methylation plays a major role in gene repression (Fig. 2f). Unexpectedly, a similar number of up- and down-regulated genes could be identified in each of the gbM, pM, and dM subsets (Fig. 2f). Thus, our results confirm the accepted repressive effect of DNA methylation on gene expression but also hint at a broad activating function of DNA methylation. Admittedly, we cannot exclude that the downregulation of methylated genes in these mutants is an indirect consequence of the upregulation of methylated genes or derives from additional side effects of the depletion of this epigenetic mark.

Interestingly, using heatmaps to compare the gene expression level of overlapping DEGs among *ddcc*, *met1-9*, and *mddcc* mutants (Fig. 2b, c, e and Supplementary Fig 3b, c), we found that DNA methylation in the methylated genes (gbM, teM, pM, and dM) affects gene expression in multiple ways. According to the behavior of DEGs in the different mutant backgrounds, we could distinguish the following effects of DNA methylation on gene expression: Redundancy, when the expression is increased/decreased only in *mddcc*; Dosage I, when the expression is mildly increased/decreased upon loss of CG methylation (in *met1-9*), and is further increased/decreased upon loss of non-CG methylation (in *mddcc*); Dosage II, when the expression is mildly increased/decreased upon loss of non-CG methylation (in *ddcc*),

and is further increased/decreased upon loss of CG methylation (in *mddcc*); Dosage III, when the expression is weakly increased/decreased upon loss of either CG or non-CG methylation (in both *met1-9* and *ddcc*) and is further increased/decreased upon complete loss of DNA methylation (in *mddcc*); Silenced/Activated by $^mCG$, when the degree of gene up- or down-regulation in *met1-9* is comparable to that in *mddcc*; and Silenced by non-$^mCG$, when the degree of gene up-regulation is similar between *ddcc* and *mddcc* (Fig. 2e, Supplementary Fig 3b, c, and Supplementary Figs. 6–8). The number of genes regulated by a Dosage effect is significantly higher than that of genes regulated by Redundancy, Silenced/Activated by $^mCG$, and Silenced by non-$^mCG$ genes (Fig. 2g), suggesting that either DNA methylation regulates gene expression mainly in a dose-dependent manner, or CG and non-CG methylation exert a redundant effect on gene expression. Compared to Dosage I regulation, which is the most common dosage effect, Dosage II and III cases are rare (Fig. 2g), indicating that CG methylation plays a major role in regulating gene expression under normal conditions, whereas the main function of non-CG methylation is to prevent misexpression of genes when CG methylation is absent. Of note, consistent with previous results[56], we found that CHG and CHH methylation of methylated DEGs in *met1-9* was mildly decreased relative to the WT. Therefore, changes in non-CG methylation could also contribute to the changes in gene expression detected in *met1-9* (Supplementary Fig. 9).

It should be noted that a substantial proportion of DEGs in *ddcc*, *met1-9*, and *mddcc* are UM genes (Fig. 2h). The expression changes in these genes could be indirectly caused by changes in the expression of methylated genes, although the possibility that the differential expression of these genes results from the loss of trace levels of DNA methylation cannot be ruled out (Fig. 2e and Supplementary Fig. 3b, c).

It has been previously shown that DNA methylation in an intron promotes the polyadenylation of the full length transcript for several genes, as exemplified by intronic DNA methylation in the *IBM1* gene[44,46,47]. Consistent with the previous results, 3′ transcripts of the *IBM1* gene are dramatically decreased in *ddcc*, *met1-9*, and *mddcc* relative to WT (Supplementary Fig. 10). To investigate the extent to which DNA methylation in introns promotes the accumulation of 3′ transcripts, we selected a subset of 173 teM genes with heavy methylation in introns, which we called Intron-teM genes (Fig. 2d). A whole gene can be divided in two parts (5′ and 3′) by a methylated intron. After excluding genes with low levels of 5′ transcripts (see the "Methods" section), we retained 52 Intron-teM genes, 9 of which showed a dramatic decrease in 3′ transcripts in *mddcc* compared to WT (Supplementary Table 3, $\mathrm{Log}_2\ [(mddcc\ 3'/5')/(\mathrm{Col}\text{-}0\ 3'/5')] < -1)$.

Intronic DNA methylation in Intron-teM genes promoting the stability of 3′ transcripts also shows various patterns of regulation, including Redundancy, when a downregulation of 3′ transcripts is found only in the complete absence of DNA methylation (Fig. 2i); Dosage, when the decrease in 3′ transcripts correlates with the quantitative loss of intronic DNA methylation (Fig. 2i); $^{m}$CG, when the degree of downregulation of 3′ transcripts is comparable between *met1-9* and *mddcc* mutants (Supplementary Fig. 10); and $^{m}$CG-and-non-$^{m}$CG, when the degree of downregulation of 3′ transcripts is comparable among *ddcc*, *met1-9*, and *mddcc* mutants (Supplementary Fig. 10). Our findings suggest that intronic DNA methylation is required for the proper expression of some Intron teM genes by promoting the production of their full-length transcripts, although the number of teM genes for which 3′ transcripts are decreased in *mddcc* is limited.

**CG and non-CG methylation jointly repress TE expression and transposition.** We identified differentially expressed TEs (DETs) in the DNA methyltransferase mutants relative to WT (DET; fold change > 2, *p*-adjusted < 0.01). As expected, most DETs are up-regulated in these mutants (Fig. 3a), and the results support that methylation in the CG context has a major role in suppressing TE expression, with non-CG methylation acting redundantly with CG methylation to repress a subset of TEs (Fig. 3a). Consistent with what is observed in genic regions, the effect of DNA methylation on TE expression displays various patterns, including Redundancy, Dosage I, Dosage II, Dosage III, Silenced by $^{m}$CG, and Silenced by non-$^{m}$CG (Fig. 3b, c and Supplementary Fig. 11). Thus, CG and non-CG methylation collaborate to repress TE expression in various ways, and there is a dose/effect relationship between DNA methylation and TE expression.

To examine whether reactivation of TEs in these mutants may result in TE transposition, we performed DNA sequencing (DNA-seq) in WT, *ddcc*, *met1-9*, and *mddcc* mutants (Supplementary Fig. 12a and Supplementary Table 4). By analyzing the DNA-seq data, 1, 1 and 12 putative TE transposition events were identified in *ddcc*, *met1-9*, and *mddcc*, respectively (Supplementary Table 5); however, only 11 TE transposition events, all in *mddcc*, could be confirmed by PCR (Fig. 3d–i and Supplementary Fig. 12). The 11 transposition events involved 5 TEs (Fig. 3j and Supplementary Table 5); these transposed TEs (AT1TE42210, AT2TE20205, AT5TE65370, and AT1TE49860) show the highest expression levels in *mddcc*, suggesting that the high degree of TE transcriptional activation only partially correlates with transpositional activation. Interestingly, we found that only a partial and antisense transcript of AT2TE42810 is increased in *mddcc* relative to *met1-9*, implying that the activation of this part of the locus may be more important than the other part for AT2TE42810 transposition (Fig. 3j). Consistent with previous results showing that mutations in *MET1* cause transposition of a limited number of TEs only after continuous selfing[35], we did not find any transposed TEs in the *met1-9* mutant, which is newly generated by CRISPR-Cas9 (Supplementary Table 5). Of note, in the *mddcc* mutant, the 11 transposition events were identified in the first homozygous generation. Our findings confirm prior results indicating that CG and non-CG methylation act redundantly to prevent TEs transposition.

**Expression of antisense transcripts and non-annotated transcripts in DNA methyltransferase-deficient mutants.** DNA methylation (both CG and non-CG) in gene bodies has been proposed to suppress the expression of intragenic antisense transcripts[57–59]; however, mutation in *MET1*, which results in a substantial loss of gene body methylation, causes overexpression of only a handful of intragenic antisense transcripts[29]. We hypothesized that CG and non-CG methylation may act

redundantly to suppress the expression of intragenic antisense transcripts. To test this idea, we identified antisense transcripts with significantly altered expression (fold change > 4, *p*-adjusted < 0.01) in *ddcc*, *met1-9*, and *mddcc* mutants compared to WT. The numbers of misexpressed antisense transcripts in *met1-9* and *mddcc* mutants are low, but higher than those in *ddcc*. The observed increase in *mddcc* relative to *met1-9* is minor (Fig. 4a, b). As expected, most of the genes with up-regulated antisense transcripts harbor heavy DNA methylation in WT plants (Fig. 4c, d). These results support a role of CG and non-CG methylation in the gene bodies in suppressing antisense transcription on a restricted number of loci.

Additionally, we searched for non-annotated transcripts with different expression levels (fold change > 2, *p*-adjusted < 0.01) in *ddcc*, *met1-9*, and *mddcc* mutants relative to WT. Interestingly, we identified 46 non-annotated transcribed elements in the *mddcc* mutant (Fig. 4e, f). The vast majority of loci of up-regulated non-annotated transcripts display dense DNA methylation in WT plants (Fig. 4g, h). This finding suggests that additional transcribed elements, whose expression is blocked by DNA methylation, exist in the *Arabidopsis* genome.

**DNA methylation is required for normal *Arabidopsis* development but dispensable for its survival.** To investigate the significance of DNA methylation for plant development, we phenotyped the *ddcc*, *met1-9*, and *mddcc* mutants throughout their life cycle. While seedling size in *ddcc* and *met1-9* is similar to that in WT (Fig. 5a), *mddcc* seedlings are extremely small (Fig. 5a). The reduction in size is even more pronounced when *mddcc* plants are grown in soil, where they display a range of phenotype severities (Fig. 5b). This phenotypic variability among *mddcc* individual plants may be caused by TE transposition in these plants (Fig. 3 and Supplementary Fig. 12). Noticeably, although *mddcc* plants grow slowly and exhibit severe developmental alterations, their survival span is longer than that of other genotypes in long-day conditions (Supplementary Fig. 13), leading to the conclusion that DNA methylation is necessary for *Arabidopsis* development but dispensable for its survival. Nevertheless, we found that the *mddcc* mutant never flowered (Fig. 5c and Supplementary Fig. 13), which correlates with the aberrant expression of genes involved in floral development (Fig. 5d). Previous studies have demonstrated that the loss of DNA methylation at the *FWA* promoter in *met1* mutants causes ectopic *FWA* expression, which results in late flowering[60]. Since the expression levels of *FWA* are comparable between *met1-9* and *mddcc* (Fig. 5e), other genes must underlie this phenotypic difference between the two mutants.

Intriguingly, mutation in *MET1* is sufficient to alter pavement cell shape from irregular (jigsaw-puzzle shape)[61] to regular (lacking interdigitation of lobes) (Fig. 5f and Supplementary Fig. 14a), and to cause spontaneous cell death (Supplementary Fig. 14c). Consistently, cytoskeleton organization-related genes with pivotal roles in controlling pavement cell shape[62,63] and cell death-related genes are sharply down-regulated and up-regulated, respectively, in both *met1-9* and *mddcc* mutants (Fig. 5d and Supplementary Fig. 14d). In addition, CG methylation affects endoreduplication in nuclei isolated from cotyledons from 11-day-old seedling, since the distribution of DNA content in nuclei from *met1* and *mddcc*, which peaks at 8 °C, is different from that in nuclei from WT and *ddcc*, which peaks at 16 °C instead (Supplementary Fig. 14b)[61]. In contrast to WT, *ddcc*, and *met1*, where trichomes generally have three branches, the majority of trichomes in *mddcc* have two branches (Fig. 5g and Supplementary Fig. 14e), indicating that CG and non-CG methylation redundantly contribute to trichome development. In addition,

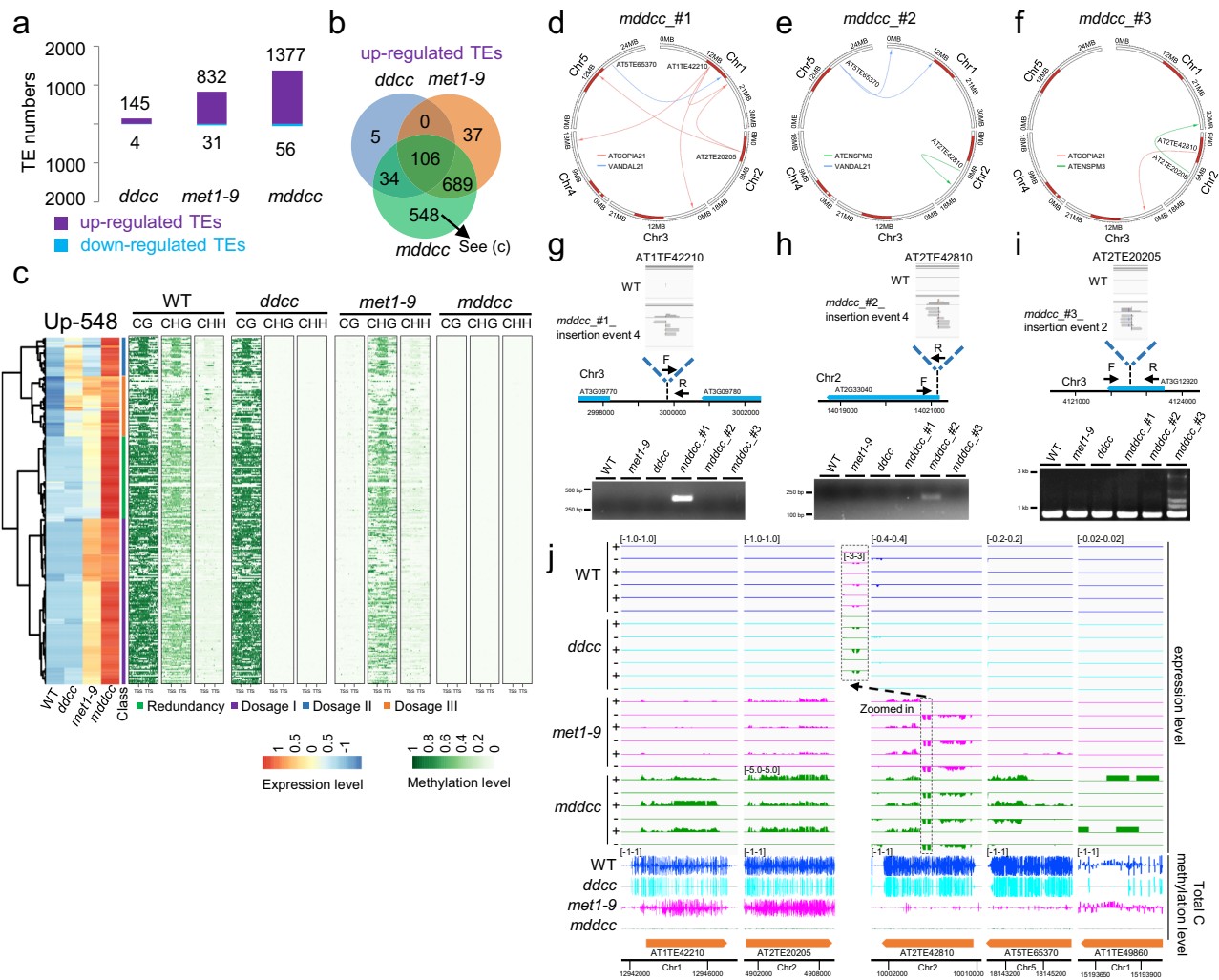

**Fig. 3 CG and non-CG methylation jointly repress TE activation and movement. a** Number of differentially expressed TEs (DETs) in the indicated mutants relative to WT. **b** Venn diagram showing the overlap among up-regulated TEs in *ddcc*, *met1-9*, and *mddcc* mutants. **c** Heat maps showing the expression and DNA methylation patterns of TEs in the indicated groups. Based on the expression patterns of TEs, TEs were classified into the following categories: Redundancy, Dosage I, Dosage II, and Dosage III (for details, see Fig. 2e). **d–f** Circle plots showing TE transposition in different *mddcc* individual plants. Arrows start and end represent the original and inserted sites of transposed TEs, respectively. The red/green/blue arrows indicate that the transposed TE belongs to the ATCOPIA21, ATENSPM3, or VANDAL21 subfamily, respectively. **g–i** Products of PCR amplification with paired primers flanking new insertion sites or a transposon-specific primer and a primer flanking the new insertion site in the indicated genotypes. Black arrows indicate primers. Upper panels: screenshots from Integrative Genomic Viewer (IGV) showing split-reads for TE insertions. Lower panels, coordinate lines indicate the sequence contexts of the new insertion sites. **j** Snapshots of expression and DNA methylation levels over transposed TEs in the indicated mutants. Source data are provided as a Source Data file.

DNA methylation is essential for proper vascular development, since vasculature is dramatically reduced in the *mddcc* mutant (Supplementary Fig. 15a–c). Gene expression analyses support our findings, since multiple genes involved in trichome differentiation and vascular development are drastically misexpressed in *mddcc* (Fig. 5d and Supplementary Fig. 15d). Given the extreme developmental defects of the *mddcc* mutant, we wondered whether DNA methylation might be required for meristem activity. While the shoot apical meristem (SAM) in both *met1-9* and *ddcc* mutants is comparable to that in WT (Supplementary Fig. 16a, b), the SAM in the *mddcc* mutant is severely reduced (Supplementary Fig. 16a, b), implying that CG and non-CG methylation are redundantly involved in the development of the SAM. In line with this observation, misregulation of genes involved in SAM development is maximal in the *mddcc* mutant (Supplementary Fig. 16c).

Root length is slightly reduced in *ddcc* and *met1* compared to the WT, but is drastically decreased in *mddcc* (Fig. 6a). To examine the activity of the RAM, we applied the thymidine analog 5-ethynyl-2′-deoxyuridine (EdU) to visualize cell proliferation in this area[64,65]. We found that the EdU signal, a proxy for DNA replication and cell division, is dramatically decreased in *met1-9*, and is virtually undetectable in *mddcc* (Fig. 6b), demonstrating that CG methylation contributes to cell proliferation in the RAM, with non-CG methylation acting partially redundantly. Remarkably, we found that xylem differentiation and root hair differentiation occur prematurely in *mddcc*, being frequently observed directly at the root tip (Fig. 6c–e and Supplementary Fig. 17); this phenotype suggests that CG and non-CG methylation act redundantly to either inhibit root cell differentiation and/or promote meristem maintenance.

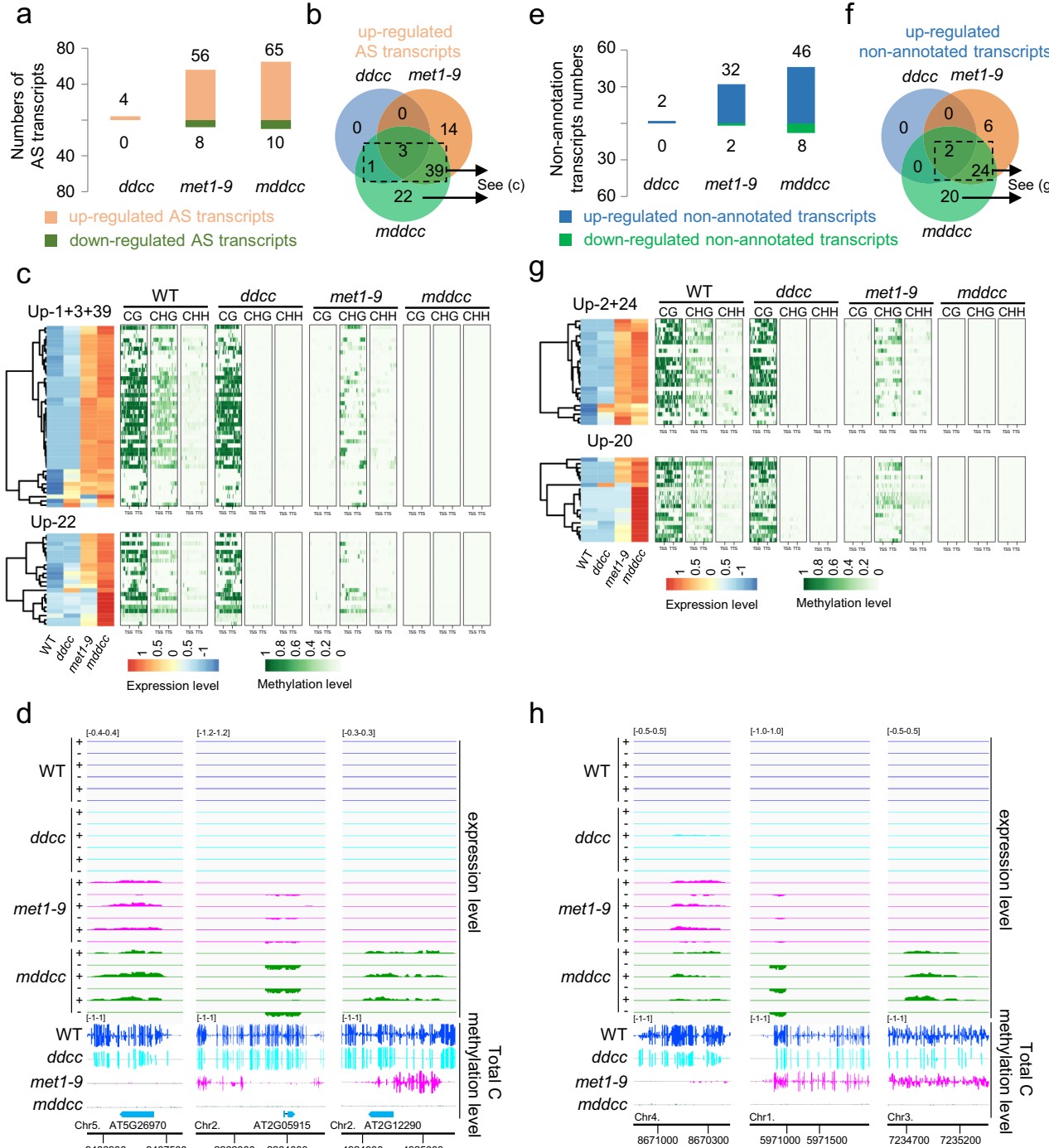

**Fig. 4 DNA methylation represses the expression of antisense and non-annotated transcripts. a** Numbers of differentially expressed antisense (AS) transcripts in the indicated mutants relative to WT. **b** Venn diagram showing the overlap among up-regulated AS transcripts in *ddcc*, *met1-9*, and *mddcc* mutants. **c** Heat maps showing the expression and DNA methylation patterns of the antisense transcripts in the indicated groups. **d** Snapshots of expression and DNA methylation levels over three antisense transcript genes in the indicated mutants. **e** Number of differentially expressed non-annotated transcripts in the indicated mutants relative to WT. **f** Venn diagram showing the overlap among up-regulated non-annotated transcripts in *ddcc*, *met1-9*, and *mddcc* mutants. **g** Heat maps showing the expression and DNA methylation patterns of the non-annotated transcripts in the indicated groups. **h** Snapshots of expression and DNA methylation levels over three non-annotated transcripts in the indicated mutants.

## Discussion

The mechanisms regulating DNA methylation in plants have been extensively studied in the model species *Arabidopsis* in the past decades[1,2,4–9,12–14,28,66]. However, DNA methylation-free mutant plants have not been available thus far, and consequently it remained unclear to what extent this epigenetic modification was required for plant growth and development. Here, we mutated all known functional DNA methyltransferases in *Arabidopsis*, generating a quintuple mutant devoid of DNA methylation (Fig. 1). Our results demonstrate that DNA methylation in *Arabidopsis* is mediated entirely by the five previously identified DNA methyltransferase enzymes. The extremely low levels of DNA methylation in *mddcc* are likely attributable to non-conversion of unmethylated cytosines (Supplementary Table 1).

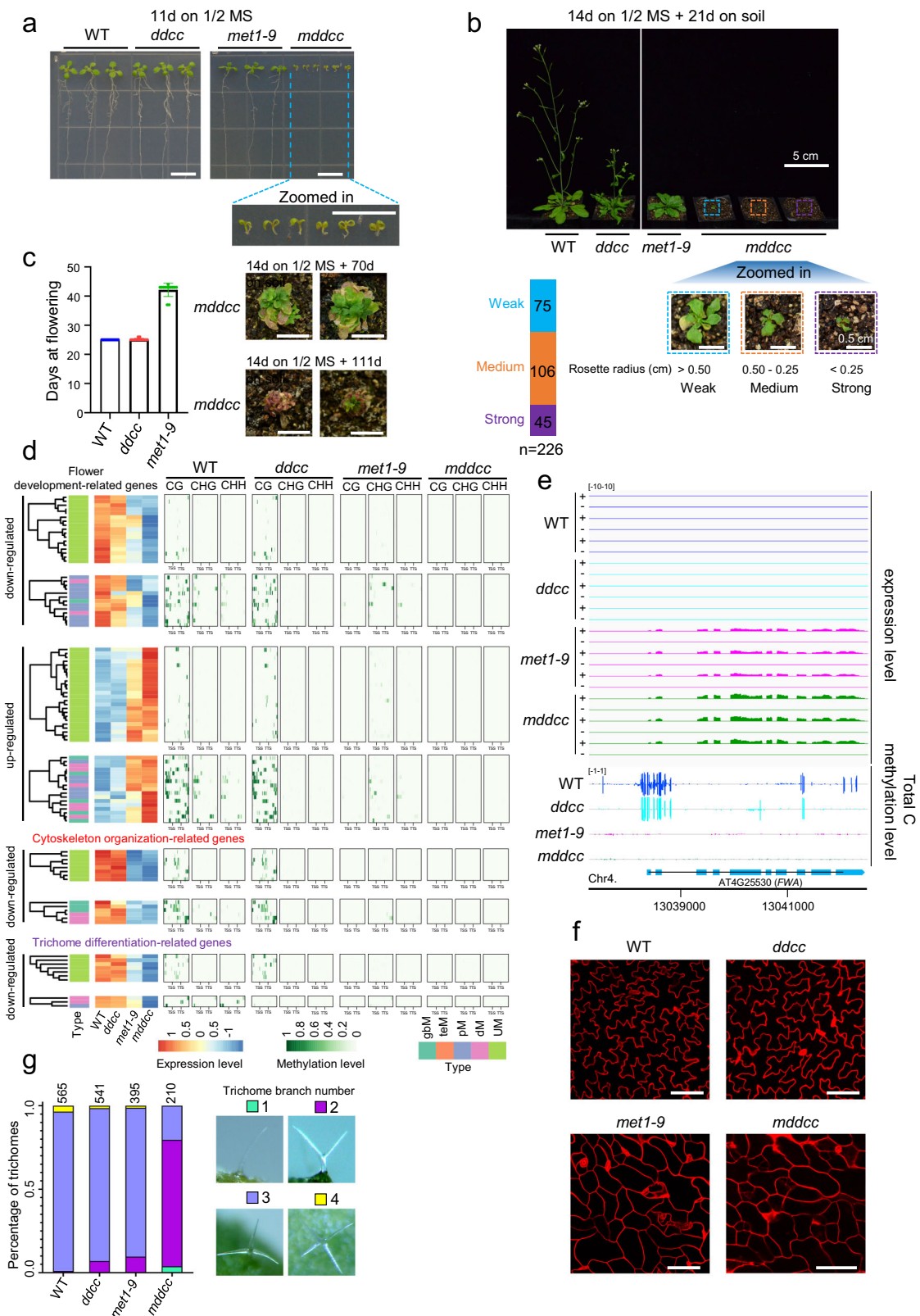

In *Arabidopsis*, there are three *MET1* homologs (AT4G14140, AT4G13610, and AT4G08990), of which the potential function in DNA methylation is unexplored. Considering that these homologous genes are expressed in seedlings[67] and that DNA methylation is completely erased in *mddcc*, it seems likely that the encoded proteins do not play a prevalent role in maintaining DNA methylation, although we cannot exclude that they may

maintain DNA methylation in endosperm since their expression levels are up-regulated in endosperm relative to other tissues[68]. Of note, a function of these or other yet-to-be-identified DNA methyltransferases in very limited specific cells or at certain developmental stages, of which the contribution might have been diluted in our experimental approach, can nevertheless not be ruled out at this point.

**Fig. 5 The *mddcc* mutant exhibits a suite of extreme developmental defects and fails to flower. a** Phenotypes of 11-day-old seedlings of the indicated genotypes on 1/2 MS media. Scale bar, 1 cm. **b** Phenotypes of 35-day-old plants of the indicated genotypes. After growing for 14 days on 1/2 MS media, the plants were transplanted into soil for 21 days of growth. Based on the rosette radius, the phenotypes of the *mddcc* mutant were classified into Weak, Medium and Strong, as indicated; the bar plot shows the relative abundance of each of these categories. **c** The *mddcc* mutant never flowers. Left, bar pot showing the flowering times of the indicated genotypes grown in long-day conditions. The data are the means ± SD of the biological repeats ($n = 13$). Right, phenotypes of 84-day-old and 125-day-old *mddcc* plants. After growing for 14 days on 1/2 MS media, the plants were transplanted into soil for 70 or 111 days of growth. Scale bar, 1 cm. **d** Heat maps showing the expression and DNA methylation patterns of flower development-related genes (upper panel), cytoskeleton organization-related genes (middle panel), and trichome differentiation-related genes (lower panel) in the indicated genotypes. Flower development-related genes (see Supplementary Data 1) were selected from the overlap between GO:0009908 (flower development) and DEGs in *mddcc*. Cytoskeleton organization-related genes (see Supplementary Data 1) were selected from microtubule-related GO terms over-represented in the subset of down-regulated genes common to *met1-9* and *mddcc* (Fig. 2c). Trichome differentiation-related genes (see Supplementary Data 1) were identified from the overlap between GO:0010026 (trichome differentiation) and down-regulated genes in *mddcc*. **e** Snapshots of expression and DNA methylation levels over AT4G25530 (*FWA*) in the indicated mutants. **f** Representative images of pavement cell morphology of cotyledons from 11-day-old WT, *ddcc*, *met1-9*, and *mddcc* seedlings. Cell outlines were visualized with PI. Scale bar: 100 μm. Experiments were independently repeated two times with similar results. **g** Trichome branch number in WT, *ddcc*, *met1-9*, and *mddcc*. The numbers above the bars indicate the total number of trichomes analyzed. Trichomes were observed from the first true leaf of 11-day-old plants. Source data are provided as a Source Data file.

It is notable that complete removal of DNA methylation in *Arabidopsis* is not lethal, rendering individuals infertile yet viable. In contrast, null mutations in single DNA methyltransferases result in seedling or embryonic lethality in rice or maize[69,70]. Compared to *Arabidopsis*, these plants have much higher contents of TEs in their genomes[71], and consequently much higher probabilities of essential genes being affected by nearby TEs and thus by DNA methylation around the TEs. Even though DNA methylation is present in the genomes of all vertebrates and plants, there are species without DNA methylation, as exemplified by *Drosophila melanogaster*, *Caenorhabditis elegans*, and *Tribolium castaneum*[13,72]. It seems that these species have evolved other mechanisms to replace the central function of DNA methylation, namely the silencing of TEs[73]. While our manuscript was under review, Liang et al. reported that an *mddcc* mutant is embryonically lethal[74]. It should be pointed out that their failure to obtain a viable *mddcc* mutant likely results from the fact that their selected *ddcc* alleles (*drm1-2 drm2-2 cmt3-11 cmt2-7*), as well as the process employed for the generation of the quintuple mutant[74] (CRISPR/Cas9-mediated mutation of *MET1* in the WT background and subsequent crossing with the *ddcc* mutant), are different from the ones used in this work (*drm1-2 drm2-2 cmt3-11 cmt2-3*; CRISPR/Cas9-mediated mutation of *MET1* directly in the *ddcc* background (see the "Methods" section)).

The DNA methylation-free *mddcc* mutant has enabled a comprehensive assessment of the full impact of DNA methylation on the regulation of gene expression, TE silencing, and plant development. Our results indicate that the degree of functional redundancy between CG methylation and non-CG methylation in the regulation of gene expression is limited (Fig. 2), and that DNA methylation dosage is determinant (Fig. 2g). Unexpectedly, we found that nearly half of the differentially expressed, methylation-associated genes (with the exception of teM genes) are down-regulated in *mddcc* (Fig. 2f), pointing to a possibly broader role of DNA methylation than previously assumed in the promotion of gene expression. Recently discovered DNA methylation reader complexes which promote gene expression support this assumption[75–79]. Also surprising is the observation that half of the DEGs in *mddcc* belong to the unmethylated (UM) category (Fig. 2h); it is conceivable that the altered expression of these genes is a byproduct of changes in the expression of DNA methylation-associated genes, or, alternatively, is regulated by distant DNA methylation through chromosomal interactions[80].

Our results also revealed that DNA methylation in introns at Intron-teM genes promotes the accumulation of full-length transcripts (Fig. 2i and Supplementary Fig. 10). This role is in agreement with the recent discovery of a protein complex that promotes distal polyadenylation of intron-methylated genes[45–49,81]. It seems likely that DNA methylation-associated heterochromatic histone modifications may recruit this protein complex to Intron-teM genes to facilitate the production of full-length transcripts.

In contrast to gbM, dM, and pM genes, the expression of most teM genes is up-regulated in *mddcc* (Fig. 2f), suggesting that DNA methylation acts to restrain gene expression when gene bodies carry heavy non-CG methylation. Across all mutants, few DEGs belong to the gbM gene category (Fig. 2f), supporting that either CG methylation in gbM genes has a limited role in regulating gene expression, or that the role of gbM in regulating gene expression is limited, and the affected transcripts did not reach the threshold to be considered DEGs in our analyses. In a recent preprint, Shahzad et al. showed that gene body methylation tends to have a modest positive effect on gene expression in natural accessions and that natural variation in this trait is associated with environmental variables, suggesting a role in adaptive evolution[59]. In mouse cells, CG methylation in gene bodies prevents cryptic transcription initiation[82]. However, results obtained in *Arabidopsis* do not support a similar function in plants[83].

Interestingly, we found that the proportion of DEGs in dM genes is similar to that in pM genes; how DNA methylation at 3′ downstream sequences regulates gene expression is poorly understood. Conceivably, DNA methylation in these regions may affect transcription termination or transcript processing, hence affecting transcript stability, or may even impact promoter function through the formation of chromatin loops[84,85].

The subset of up-regulated genes in the DNA methylation-deficient mutants is highly enriched in defense-related GO terms (Supplementary Fig. 5a, c), which suggests a general role of DNA methylation in this process. This is consistent with recent studies showing that DNA demethylases are required for the expression of defense-related genes and for plant resistance against fungal and bacterial pathogens[86–89].

It should be noted that the age of the plants used for the transcriptomic analysis is genotype-dependent: while 2-week-old plants were used for the WT, *met1*, and *ddcc*, 5-week-old plants were used for the quintuple *mddcc* mutant. The justification for this experimental design lies on the extremely slow pace of growth and development exhibited by the latter, which renders 5-week-old *mddcc* plants more similar to 2-week-old plants of the other genotypes analyzed. To exclude the possibility that DEGs identified in *mddcc* in this study are mainly caused by the difference in sampling time, we compared the expression levels of these DEGs between 5-week-old *mddcc* and 5-week-old WT

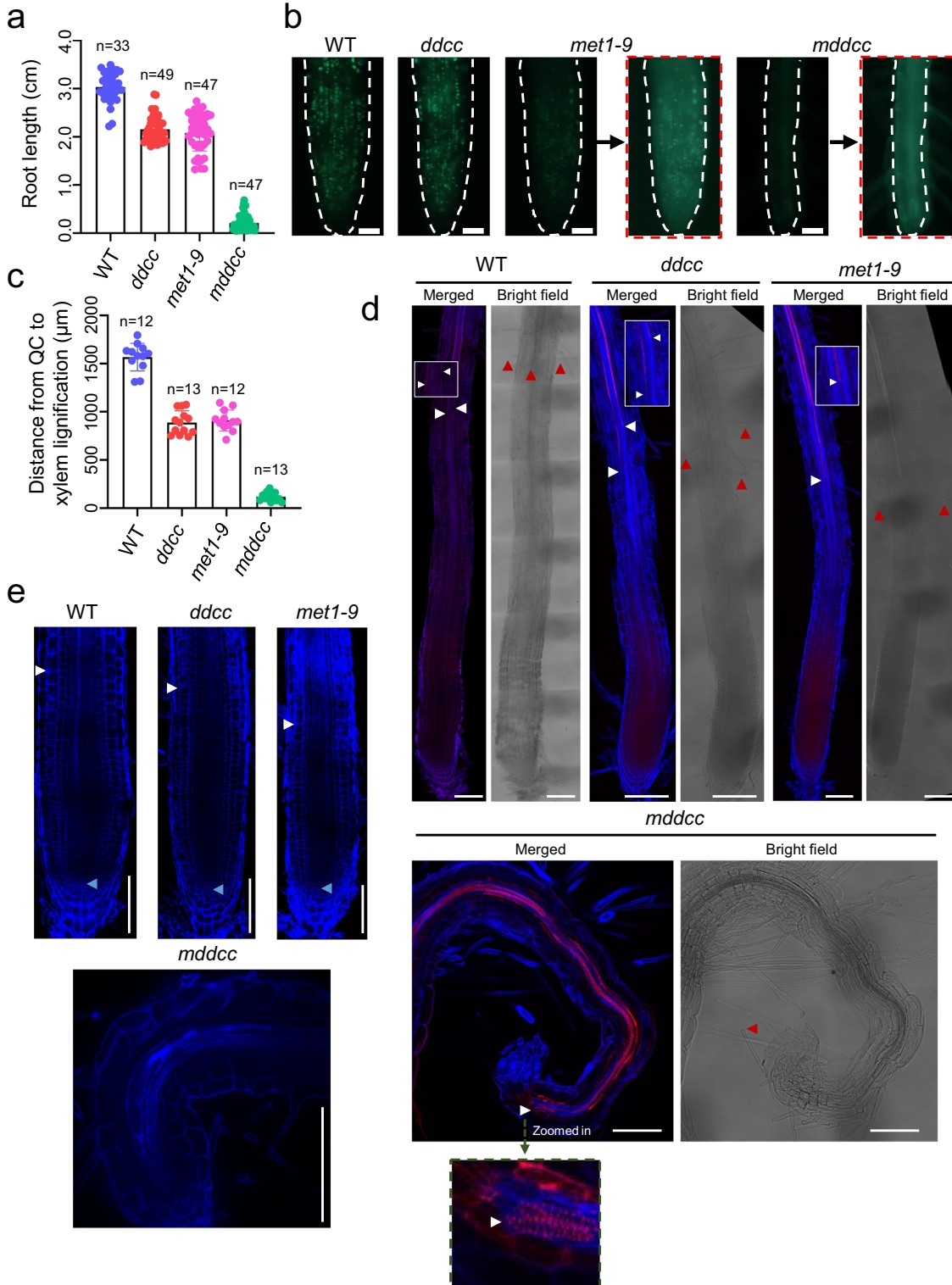

**Fig. 6 CG and non-CG methylation jointly regulate the activity of RAM. a** Quantification of root length of 9-day-old WT, *ddcc*, *met1-9*, and *mddcc* seedlings. The data are the means ± SD of the biological repeats. **b** Representative images of EdU labeling in the RAM of roots of 11-day-old WT, *ddcc*, *met1-9*, and *mddcc* seedlings. Images framed with a red dotted line show a higher exposure version of the images on the left. Scale bar: 50 μm. Experiments were independently repeated two times with similar results. **c** Distance from the QC to xylem lignification in roots of 9-day-old WT, *ddcc*, *met1-9*, and *mddcc* mutants. The data are the means ± SD of the biological repeats. **d** Cleared roots stained with calcofluor white (blue) and basic fuchsin (red). White arrowheads indicate the site of appearance of the first protoxylem cells; red arrowheads indicate root hairs. Scale bar: 100 μm. Experiments were independently repeated two times with similar results. **e** Representative images of root meristems from 9-day-old WT, *ddcc*, *met1-9*, and *mddcc* seedlings. Cell outlines were visualized with calcofluor white. Blue and white arrowheads indicate the QC and the cortex transition boundary, respectively. Scale bars: 100 μm. Experiments were independently repeated two times with similar results. Source data are provided as a Source Data file.

plants. We found that more than half of these DEGs are similarly up/down-regulated in 5-week-old *mddcc* compared to 5-week-old WT plants (Supplementary Fig. 19). Most of those DEGs that are not similarly up/down-regulated in 5-week-old *mddcc* compared to 5-week-old WT plants display the corresponding changes in 2-week-old *met1* plants relative to 2-week-old wild-type plants (Supplementary Fig. 19), suggesting that the expression of these genes can be regulated by DNA methylation rather than purely developmental stage-dependent. In parallel, we compared the expression levels of selected previously described DEGs potentially involved in specific phenotypes of the *mddcc* mutant, finding that the majority of very highly up-regulated DEGs are also up-regulated in 5-week-old *mddcc* relative to 5-week-old WT plants, despite the differences in developmental stage (Supplementary Fig. 20). Taken together, these results indicate that the transcript differences identified between the *mddcc* mutant and WT are, at least to a large extent, not artifactual due to the age difference. Nevertheless, an effect of the sampling time on the transcript differences observed cannot be excluded. Moreover, the difference in the development of *mddcc* mutant compared to other genotypes could have unintended secondary effects, which must be considered when interpreting the results. Of note, the analysis of expression of TEs, antisense, and non-annotated transcripts may be also affected by the factors mentioned above (sampling time and developmental stage).

Our results indicate that non-CG methylation fully compensates the loss of CG methylation in terms of transposition, even though this compensation is only partial in terms of TE expression. Interestingly, RdDM has been shown to preferentially target the extremities of long TEs (e.g. ONSEN and other COPIA LTR-retroelements), which is sufficient to prevent their transposition even when they are transcriptionally active[90,91]. While 12 TE subfamilies were identified as mobilized in a *ddm1*-derived epi-RIL population[38], only three of them were found as transposing in this work; nevertheless, it is worth noting that these three subfamilies account for 98% of the transposition events in the *ddm1*-derived epi-RIL. The detection of additional, lowly represented mobilized TE subfamilies in the *mddcc* mutant might have been hampered by the limited number of individuals included in the analyses. TE transposition occurs in the first homozygous generation of *mddcc*, providing direct evidence that DNA methylation plays a crucial role in protecting genome stability. DNA methylation is known to strongly impact meiotic recombination frequencies[92–94]. It would be of interest to investigate in the future whether the DNA methylation-free mutants may exhibit changes in genomic structural rearrangements.

Choi et al. proposed that DNA methylation and other factors jointly suppress the expression of antisense transcripts[95]; however, according to our results, DNA methylation would have a strong contribution in this process, yet on a limited number of loci (Fig. 4a–d). Interestingly, we found that dozens of non-annotated transcripts are expressed in the *mddcc* mutant (Fig. 4e–h). These non-annotated transcripts may represent unidentified genes or non-coding RNAs normally masked by DNA methylation. Whether these putative novel transcripts have biological functions in plants is at this point an open question.

Strikingly, the *mddcc* mutant displays extremely reduced size and a number of obvious developmental defects, fails to flower (Fig. 5 and Supplementary Fig. 13), and both the SAM and RAM and the formation of the vasculature are dramatically affected in these plants (Fig. 6 and Supplementary Figs. 15 and 16). Our results therefore suggest that meristem activity is redundantly regulated by CG and non-CG methylation. This notion is supported by the fact that DNA methylation dynamically changes during SAM development[96–98]. In contrast, pavement cell shape, cell death, and endoreduplication are specifically controlled by

CG methylation (Fig. 5f and Supplementary Fig. 14a–c). Absence of both CG and non-CG methylation in the *mddcc* mutant causes premature cell differentiation in roots; a role of this epigenetic modification in cell differentiation is consistent with the finding that the RAM displays widespread cell type-specific patterns of DNA methylation[99]. Loci encoding crucial regulators of some of the biological processes severely affected in the DNA methyltransferases-deficient mutants analyzed in this work show altered methylation patterns and gene expression levels (Supplementary Data 1). For example, the positive regulator of cell death *ACCELERATED CELL DEATH 6* (*ACD6*) loses DNA methylation in its promoter in *met1* and *mddcc*, and is concomitantly up-regulated (Supplementary Fig. 18). It has been demonstrated that over-expression of *ACD6* causes spontaneous cell death[100]. Therefore, the CG methylation-dependent cell death regulation (Supplementary Fig. 14c) could potentially function, at least in part, via modulating *ACD6* expression. Loss of methylation in the promoter of *E2Fc*, encoding a core inhibitor of cell division[101,102], correlates with a substantial increase in its expression (Supplementary Fig. 18), which may underlie the observed defects in cell proliferation in *met1-9* and *mddcc* (Fig. 6b).

In summary, our results demonstrate that CG and non-CG methylation act together to protect genome stability and regulate gene expression, ultimately controlling a suite of important developmental processes and reproduction.

## Methods

**Plant materials and growth conditions**. All plants were grown under long-day condition (16 h light/8 h dark). For seedling growth, *Arabidopsis* seeds were plated on 1/2 Murashige and Skoog (MS) medium with 0.6% agar, 0.7% agar, 1.0% or 1.2% agar and 1.5% sucrose and stratified for 7 days at 4 °C in darkness before being transferred to the growth chamber (16 h light/8 h dark, 22 °C). For experiments with adult plants, 14-day-old seedlings were transplanted to soil in the growth chamber. All mutant lines used in this study are in the Columbia-0 (Col-0) background. The *ddcc* mutant used in this study was generated by genetic crossing (*cmt2 × ddc*) and subsequent PCR-based genotyping in F2 populations. The *mddcc* and *met1-9* mutants used in this study were generated by CRISPR/Cas9 system in *ddcc* and WT background, respectively (see below).

**Generation of *ddcc*, *met1-9*, and *mddcc* mutants**. We crossed the *cmt2*[20] mutant to the *ddc*[103] triple mutant and obtained the *ddcc* mutant by genotyping the F2 populations. Primer sequences are listed in Supplementary Data 2.

The constructs for CRISPR-Cas9-mediated genome editing were designed as previously described[55]. Briefly, the expression of Cas9 was controlled by the *UBQ1* promoter. The *UBQ1* terminator was placed at the end of Cas9 ORF. The nuclear localization signal (NLS) was fused to both the N and C termini of Cas9. The sgRNAs were driven by Pol III-dependent gene promoters, including U6, U3b, and 7SL. Two sgRNAs were designed to target the first exon of the *MET1* gene. The CRISPR-Cas9 construct was transformed into WT and the *ddcc* mutant backgrounds. The T1 transformants were analyzed by sequencing the sgRNAs target regions in the *MET1* gene, which were amplified by PCR using the primers listed in Supplementary Data 2. Then, we isolated homozygous *met1-9* mutants without the CRISPR-Cas9 constructs from the mutated T2 populations in the WT background. Heterozygous *met1* mutants (*met1/+*) without the CRISPR-Cas9 construct were isolated from the mutated T2 populations in the *ddcc* background. The *mddcc* mutant was isolated by genotyping the progenies of *met1/+ drm1 drm2 cmt3 cmt2* plants. Primer sequences are listed in Supplementary Data 2.

**PCR assay**. To confirm new transposon insertions, PCR was performed with a transposon-specific primer and a primer flanking the new insertion or with two primers flanking the new insertion. The DNeasy Plant Mini Kit (Qiagen) was used for DNA extraction. All PCR reactions were carried out using Ex-Taq enzyme (Takara). Primer sequences are listed in Supplementary Data 2.

**Real-time quantitative RT-PCR**. For real-time RT-PCR analysis, total RNA was subjected to reverse transcription using the TransScript One-Step gDNA Removal and cDNA Synthesis SuperMix kit (TransGen Biotech). The cDNA was used as template in a PCR reaction with Green Premix Ex Taq (Tli RNaseH Plus) (TaKaRa). All the reactions were carried out on a CFX96TM Real-Time System (Bio-Rad). The constitutively expressed EF1α was used as an endogenous control for normalization. Primer sequences are listed in Supplementary Data 2.

**Flow cytometry**. Cotyledons of 11-day-old plants were chopped with a razor blade in 1 mL Galbraith's buffer[104] (45 mM MgCl$_2$, 30 mM sodium citrate, 20 mM MOPS, 0.1% (v/v) Triton x-100, pH = 7.0). The lysate was filtered through a 40-μm cell strainer (BD Falcon) and incubated at 4 °C for 5 min. The eluate was transferred to a 15 mL tube, 2 μl DAPI solution (1 mg/mL) were added, and the mixture was incubated on ice for 10 min. Then, ploidy was analyzed by using a BD FAS-CAria III flow cytometer. The FACS data were analyzed by FlowJo 7.6. The Flow cytometry gating strategy is shown in Supplementary Fig. 21.

**Confocal microscopy**. For visualizing pavement cells, cotyledons from 4-day-old or 11-day-old seedlings were detached and incubated with propidium iodide (PI) solution (10 μg/mL) for 10 min. Then, samples were imaged by confocal microscopy (LEICA SMD FLCS). PI was excited with a 538 nm laser; emission was detected between 600 and 640 nm.

**Histological analysis**. 11-day-old seedlings or cotyledon and hypocotyl of 11-day-old pants were infiltrated with FAA solution (50% (v/v) ethanol, 5% (v/v) acetic acid, 3.7% formaldehyde) through vacuum infiltration for 15 min, and incubated with this FAA solution overnight at 4 °C (12–16 h). After fixation, samples were dehydrated in a graded ethanol series (1 h 50% ethanol, 1 h 60% ethanol, 1 h 70% ethanol); samples were then stored in 70% ethanol overnight. Next, samples were again dehydrated and infiltrated with Histo-Clear II (HS-202) according the following steps: 85% ethanol for 1 h at 4 °C; 95% ethanol for 1 h at 4 °C; 100% ethanol for 30 min at room temperature (RT); 100% ethanol for 30 min at RT; 100% ethanol for 1 h at RT; 100% ethanol for 1 h at RT; 25% Histo-Clear II/75% ethanol for 30 min at RT; 50% Histo-Clear II/50% ethanol with eosin (1.25%) for 30 min at RT; 75% Histo-Clear II/25% ethanol for 30 min at RT; 100% Histo-Clear II for 1 h at RT; 100% Histo-Clear II for 1 h at RT; 100% Histo-Clear II with 1/4 volume paraffin (REF. 39601095, Leica) overnight at RT. In the next 4 days, the samples were repeatedly infiltrated with paraffin, and then embedded. A series of 6 μm-thick longitudinal sections were made with a Leica RM 2235 microtome. Sections were transferred to microscopic slides, stained for 15 min in 0.1% toluidine blue solution, and rinsed with water. The slides were sealed by neutral balsam and visualized under a microscope (BX53F, Olympus).

**Examination of cell death**. Cell death was determined by trypan blue staining, following a published method[105]. Plant cotyledons were placed in TB staining solution (0.02 g of trypan blue and 10 g of phenol dissolved in 30 mL of mixed solution (1/3 (v/v) glycerol, 1/3 (v/v) lactic acid, 1/3 (v/v) water); this solution was further diluted with ethanol in 1:2 (v/v) and boiled for 2 min. Then, the tubes were placed in the fume hood for 1 h with gentle shaking at RT. Then, the nonspecific staining was removed with the destaining solution (250% (m/v) chloral hydrate, pH = 1.2). Plant tissues were then kept in 10% (v/v) glycerol for imaging. Imaging was done using an SZX7 microscope (OLYMPUS).

**Analysis of trichomes**. The first true leaf of 11-day-old plants were selected to do this assay. Quantification of trichome branching was done manually by counting the number of branches under the microscopy (SZX7, Olympus). At least 23 plants per genotypes were observed.

**Imaging leaf vein**. Leaf vein patterning was imaging according to a previously described procedure with minor modifications[106]. Briefly, cotyledons were fixed in 100% ethanol:acetic acid (6:1, v/v) overnight at 4 °C. They were then washed once in 100% ethanol and again in 70% (v/v) ethanol, followed by clearing in chloral hydrate solution (chloral hydrate/glycerol/water (8:1:3)) for 1 h at RT. Then, cotyledons were rinsed twice in water. Cotyledons were cleared again in 85% (w/v) lactic acid for at least 3 days at RT. Cotyledons were placed in lactic acid and observed using IX73 microscopy (OLYMPUS).

**Root imaging**. Root apices of 9-day-old seedlings were fixed with 4% paraformaldehyde, treated with ClearSee solution[107], and stained with calcofluor-white and basic fuchsin. Confocal images were taken with a Leica TCS SP8 point scanning confocal microscope with the following settings: for calcofluor-white, Ex: 405 nm, Em: 425-475 nm; for basic fuchsin, Ex: 561 nm, Em: 600–650 nm. ImageJ was used for imaging analysis.

**5-ethynyl-2′-deoxyuridine staining**. 11-day-old seedlings were incubated with 5-ethynyl-2′-deoxyuridine (EdU) (1 μM) (Cat#C10350, Invitrogen) for 30 min in liquid 1/2 MS medium. Then, samples were fixed with 1× PBS solution containing 4% formaldehyde and 0.1% Triton x-100 for 30 min at 4 °C. After fixation, samples were washed with 1× PBS 3 times with rotation for 5 min at RT, then conjugated to Alexa Fluor 488 with the Click-iT EdU Alexa Fluor 488 HCS Assay (Cat#C10350, Invitrogen) for 30 min in the dark at RT. Samples were then washed with 1× PBS 3 times with rotation for 10 min at RT and imaged using a stereomicroscope (Leica DM6B).

**DNA-seq and identification of non-reference TE insertions**. The DNeasy Plant Mini Kit (Qiagen) was used for DNA extraction. Library construction and sequencing were performed at the PSC Genomics Core Facility. Identification of non-reference TE insertions with target site duplications (TSDs) was conducted using SPLITREADER with some modifications[108]. In brief, after trimming low-quality sequences and adapters using Trimmomatic, clean read pairs were mapped to the reference genome using Bowtie2 with the parameter "-very-sensitive". Subsequently, unmapped reads from both pairs, including discordantly mapped reads, were extracted and merged together. Those unmapped reads were remapped to a collection of 5′ and 3′ TE extremities (300 bp) sequence with parameters "–local–very-sensitive" (TE families like ARNOLDY2, ATCOPIA62, ATCOPIA95, TA12, and TAG1 families were excluded, since they do not contain copies with intact extremities in the TAIR10 reference genome[108]) and reads with soft-clipped mapping (with one end ≥20 nt mapped to the TE extremity) were selected. Those selected reads were further recursively soft-clipped by 1 nt and mapped to the reference genome using Bowtie2 with parameters "–mp 13–rdg 8,5–rfg 8,5 --local --very-sensitive" until the soft-clipped read length reached 20 nt. For reads simultaneously clipped-mapped to the TE reference and the reference genome, we further require that the other pair of the clipped read was also mapped and met one of the following criteria: (a) the other pair was properly mapped (insertion size <3000 bp and on the opposite strand) on the reference genome; (b) the other pair was properly mapped on the same TE reference; (c) the other pair is also clipped-mapped for the same TE reference and reference genome on the opposite strand. Around the TSD insertion sites, read clusters composed of four or more reads clipped from the same extremity and overlapping with read clusters composed of reads clipped from the other extremity were taken to indicate the presence of a bona fide TE insertion only if the size of the overlap was more than 3 bp and <20 bp. Putative non-reference TE insertions overlapping with aberrant genomic regions (3 kb away from the centromeric region based on Repbase annotation[109], 3 kb away from the extremities of each chromosome and regions within 500 bp of "NNNN" sequence) or spanning the corresponding donor TE sequence were filtered out.

**Whole-genome bisulfite sequencing and analysis**. Genomic DNA was extracted from the aerial part of 2-week-old (WT, *met1-9*, and *ddcc*) or 5-week-old (*mddcc*) plants using DNeasy Plant Maxi Kit (Qiagen). Bisulfite treatment (EpiTect Plus Bisulfite Kits, Qiagen), library construction (Ultra II DNA Library Prep Kit, NEB), and sequencing (Illumina Hiseq x10) were performed at the PSC Genomics Core Facility.

For data analysis, reads containing adapters and low-quality reads (*q* < 20) were trimmed using cutadapt[110] and Trimmomatic[111], respectively, and clean reads that were shorter than 45nt were discarded. The remaining clean reads were mapped to the *Arabidopsis* TAIR 10 genome using BSMAP(2.90)[112] with default parameters. In order to reduce the effect of RNA-DNA hybrids that interfere with bisulfite treatment, the reads were also mapped to the genome using bowtie2, and the reads with 0/1 mismatch both in BSAMP and bowite2 were filtered. Then methratio.py script was used to extract methylation ratio from filtered mapping results; the option -r was used to remove potential PCR duplicates. Cytosine positions with at least 4 reads coverage were retained for further analysis.

**Defining DNA methylation-associated genes**. Genes were classified into six groups using a modified version of the binomial test[113]. This approach tests for enrichment of C, CG, CHG, and CHH against a background level calculated from the whole genome. The total number of C counts and the total number of C+T counts within gene bodies (transcribed regions) of each gene were computed. The total number of C and C+T at all cytosine positions within each 500 bp bin of each gene promoter (2 kb) and downstream region (2 kb) were also computed, and the region with maximum DNA methylation level was selected for further analysis. A one-tailed binomial test was then applied to each gene for each context testing against the background methylation level in gene body, gene promoter, and gene downstream region, respectively. To control for false positives at the extremes, *q* values were calculated from *P* values by adjusting for multiple testing using the Benjamini–Hochberg false discovery rate.

gbM genes were defined by a gene body with a CG methylation *q* value <0.01 and CHG and CHH methylation *q* values ≥ 0.01. teM genes were defined by CHG or CHH methylation *q* values < 0.01. According to the CHG or CHH methylation *q* values < 0.01 of exon or intron, teM genes were divided into Intron teM genes and Other-teM genes. In addition to gbM and teM genes, genes could be classified into pM genes and dM genes. pM genes were defined by a gene promoter with a C methylation *q* value < 0.01. dM genes were defined by a downstream region with a C methylation *q* value < 0.01 and promoter with a C methylation *q* value ≥ 0.01. Other genes were classified as unmethylated (UM) genes. You can find the categories of genes in the Supplementary Data 3.

**mRNA-sequencing data analysis**. Total RNA was extracted from the aerial part of 2-week-old WT, *met1-9*, and *ddcc*, or 5-week-old WT (removing the inflorescence) and *mddcc* using RNeasy Plant Mini Kit (Qiagen) and RNase-Free DNase Set (Qiagen). Library construction (RiboMinus Plant Kit, Invitrogen; Ultra II

Directional RNA Library Prep Kit, NEB) and sequencing (Illumina Hiseq x10) were performed at the PSC Genomics Core Facility.

For RNA-seq data processing, quality control was performed using FastQC (www.bioinformatics.babraham.ac.uk/projects/fastqc). RNA-Seq reads were trimmed using cutadapt[110] and Trimmomatic[111] before alignment. The trimmed reads were aligned to the *Arabidposis* TAIR10 genome using STAR (v2.5.3)[114] with parameters "--outSAMmultNmax 1" and "--outSAMstrandField intronMotif" for running Cufflinks to assemble. The tool htseq-count of Python package HTSeq[115] was used to count the mapped fragments for each gene and TE with "--stranded=reverse" for differentially expressed genes analysis and "--stranded = yes" for differentially expressed antisense transcription of genes.

In order to identify non-annotated transcripts, cufflinks was used to assemble transcriptomes from the results of STAR. Cuffmerge were used to merge all assemblies into a master transcriptome, which was compared to known gene transcripts. The transcripts with "class_code = u" were selected as non-annotated transcripts. The non-annotated transcripts overlapping with transposons elements were removed. The tool htseq-count of Python package HTSeq[115] was used to count the mapped fragments for each non-annotated transcript with "--stranded = reverse" for differentially expressed non-annotated transcripts analysis.

The output count table was used as the input for DESeq2[116] to compute the differentially expressed genes/TE/antisense transcripts/non-annotated transcripts.

To identify Intron-teM genes with down-regulated expression downstream of their methylated intron in mutants relative to WT, the gene region upstream of the intron with the maximum non-CG methylation level was designated as the 5′ part (5′), the gene region downstream of the intron was designated as the 3′ part (3′), and the numbers of mapped reads in 5′ and 3′ were calculated using featureCounts with parameters -M -p -s 2, separately. The Intron-teM genes in which the 5′ reads in WT were ≥10, were retained for the following analysis. The reads ratio of 3′/5′ was used to monitor expression changes in the 3′ of Intron-teM genes, and the 3′ down-regulated Intron-teM genes were determined using $Log_2[(3′/5′$ counts ratio in mutant$)/(3′/5′$ counts ratio in WT$)] < −1$. Using this approach, we obtained 52 Intron-teM genes, 10 of which showed 3′ down-regulation in these mutants compared with the WT (Supplementary Table 3).

**GO-enrichment analysis**. GO-enrichment analysis of genes was performed using agriGO v.2[117] (http://systemsbiology.cau.edu.cn/agriGOv2/classification_analysis.php?category=Plant&&family=Brassicaceae).

**Reporting summary**. Further information on research design is available in the Nature Research Reporting Summary linked to this article.

## Data availability

All data supporting the findings of this study are available within the manuscript and its supplementary files. All high-throughput sequencing data generated in this study have been deposited in GEO with accessions codes GSE169497. Mutant seeds used to isolate the *mddcc* quintuple mutant are available at the Arabidopsis Biological Resource Center (ABRC) (stock numbers CS72761). Source data are provided with this paper.

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

## Acknowledgements

We thank the Core Facility for Genomics and Plant Cell biology at Shanghai Center for Plant Stress Biology (PSC), Chinese Academy of Sciences, for technical support. This work was supported by the National Natural Science Foundation of China (32100458 to L.H.) and the Chinese Academy of Sciences (to J.-K.Z.).

## Author contributions

L.H. and J.-K.Z. designed the research. L.H. performed most experiments. H.H. carried out bioinformatics analyses, with the exception of TE insertion analysis. M.B. performed the root-related assays. L.H. and Y.Y. performed trichome-related assay. C.Z. performed TE insertion analysis. J.M. performed EdU-labeling assay. L.H. and L.Z. performed RNA sequencing. L.H., R.L.-D., and J.-K.Z analyzed the data and wrote the manuscript.

## Competing interests

The authors declare no competing interests.
