## [Peer Review File · Nature Communications]

DNA methylation-free Arabidopsis reveals crucial roles of DNA methylation in regulating gene expression and developmentEditorial Note: This manuscript has been previously reviewed at another journal that is not operating a transparent peer review scheme. This document only contains reviewer comments and rebuttal letters for versions considered at Nature Communications.

Reviewers' Comments:

Reviewer #1:

Remarks to the Author:

The manuscript submitted to Nature Communication by He et al., "DNA-methylation-free Arabidopsis reveals crucial roles of DNA methylation in regulating gene expression and plant development", is a revised version of a manuscript originally submitted to Nature Genetics. The authors have considered most of the concern I expressed in my previous review and answered to some questions raised. I found the manuscript and especially the figures improved. Still, the overstated claim for novelty, also noticed by the other reviewers, is not completely corrected. The increase in biological understanding of the role of DNA methylation is limited and the interpretation of the data regarding gene expression and development restricted. This requires at least a change in the abstract. Although modified, its last sentence is still promising something that the manuscript does not hold: there is no "new insights into the function of this genomic modification in higher eukaryotes". This part of the sentence should be removed.

Minor: The revised sentence in l. 248 ff., "Overall, even though the number of teM genes of which 3' transcripts are decreased in mddcc is limited, our findings support that intronic DNA methylation is required for the proper expression of some Intron teM genes by promoting the production of its full length transcript" is really bad. Change into "Our findings suggest that intronic DNA methylation is required for the proper expression of some Intron teM genes by promoting the production of its full-length transcript, although the number of teM genes, for which 3' transcripts are decreased in mddcc, is limited."

Reviewer #5:

Remarks to the Author:

I have carefully reviewed the revised manuscript and found that authors were able to address many of the reviewers' comments, through rewording and clarifying confusing or misleading statements. One critical issue remains regarding the use of 5-week-old mddcc plant for RNA-seq comparison with 2-week-old wt, met1, ddcc plants. The suggestion by reviewer #3 for better normalization with spike-in mRNA controls was not properly addressed. The purpose of spike-in control relative to same # of input cells was to address the assumption that each cell generates same amount of total RNAs. The authors should review this important paper on why spike-in is critical for global gene expression analysis (Lovén et al (2012) Cell 151 (3), 476-482). The authors used qRT-PCR of selected 25 up- and down-regulated genes plus the use of DESeq2 for normalization would be satisfactory if each cell from different genotype does produce same amount of RNAs, which may well be true but is an unproven assumption.

There is another important issue that needs to be addressed, i.e., the use of different age materials for wt/met1/ddcc (2wo) and mddcc (5wo). Since several major conclusions were drawn from transcriptome results, simply includes a disclaimer in the discussion that "this difference in sampling time might have unintended, secondary effects, which must be considered when interpreting the results" (Lines 435-436) is unacceptable. The authors have to do more to demonstrate mddcc DEGs are induced in WT at older age so that their expression change can truly be ascribed to the loss of

DNA methylation. I understand the difficulty of such experiments. I think one possible way is to compare transcriptomes of wt plants between 2W and 5W of age, even just the use of rosette leaves would be a good start. The authors should also perform 2W-2W (wt-mut) and 2W-5W (mut-mut) comparisons to filter out genes whose expression changes over development.

The authors should do a better job stating their results around the novelty of the genetic material rather than the novelty of their results to prevent criticism of over exaggeration of the significance of their findings. I have pointed out some wording suggestions below, along with specific comments of the manuscript that I hope can help during their revision.

Specific comments:

mddcc vs ddcc, were the ddcc plants derived from m/+ddcc sergeants? This was not clear in the main text and M&M section.

Lines 33-35 "however, the extent of involvement of this epigenetic mark in genome-wide gene regulation and plant developmental control is unclear." The word "unclear" is not appropriate. This gives a false sense of not much is known. This sentence should be rephrased to reflect the true nature of this manuscript, i.e., the effect of genome-wide gene regulation and plant development in a methylation-free environment.

Line 38-39, "...CG methylation and non-CG methylation alone or jointly are required for a plethora..." the work alone should be removed.

Line 117-120, suggest changing to something like "Our findings extend prior knowledges on the importance of DNA methylation for global gene expression and development....."

Line 164 impact of CG methylation on gene transcription directly and indirectly....

Line 169, the qPCR validation for up-regulate

M&M Defining DNA methylation-associated genes. The description was confusing regarding whether promoter methylation (pM genes) was calculated using 2kb or just 500-bp. (lines 641-642)

Fig 2e, bottom panel, these are the UM genes but how come they have CG methylation in WT and in ddcc? Did I miss anything? This is also seen in Fig S3b,c bottom panels.

Lines 274, AT2TE42810...., shouldn't this be AT5TE65370 instead? As AT2TE42810 expression shows no difference between met1 and mddcc.

Line 280-281, "Our findings demonstrate that CG and non-CG methylation act redundantly to prevent TE transposition". Such statement sounded as if this is a new discovery but in fact this is at best an independent confirmation for many prior results. This should be restated accordingly.

Lines 331-333, what tissue was used for flow cytometry study? The M&M said 11-d-old cotyledon, but the main text says mesophyll cell? Please clarify. If the authors can compare 11-day old nuclei, why can't they compare transcriptome of the same stage of these mutants?

Lines 371-, regarding the 3 other MET1 orthologs, their expressions are very low during development but are somewhat induced during reproduction, particularly during endosperm development (see Hsieh et al, PNAS, 2011, PMID: 21257907).

Lines 394-397, the authors should discuss recent papers on DNA methylation reader complex) PMID: 30523112, 34185870, 26400170, 30589221)

Line 400-401, "... alternatively, is regulated by distant DNA methylation through chromosomal interaction" This does not make much sense to me. Please elaborate.

Point-by-Point response to reviewers

We would like to thank the reviewers for their careful review of our manuscript. We sincerely believe that we have improved our manuscript owing to their comments and suggestions. A point-by-point response to the reviewers' comments is provided below.

Reviewer #1 (Remarks to the Author):

The manuscript submitted to Nature Communication by He et al., "DNA-methylation-free Arabidopsis reveals crucial roles of DNA methylation in regulating gene expression and plant development", is a revised version of a manuscript originally submitted to Nature Genetics. The authors have considered most of the concern I expressed in my previous review and answered to some questions raised. I found the manuscript and especially the figures improved. Still, the overstated claim for novelty, also noticed by the other reviewers, is not completely corrected. The increase in biological understanding of the role of DNA methylation is limited and the interpretation of the data regarding gene expression and development restricted. This requires at least a change in the abstract. Although modified, its last sentence is still promising something that the manuscript does not hold: there is no "new insights into the function of this genomic modification in higher eukaryotes". This part of the sentence should be removed.

Response: Following the reviewer's advice, we have now removed this statement in the Abstract section (Lines 47-48).

Minor: The revised sentence in l. 248 ff., "Overall, even though the number of teM genes of which 3' transcripts are decreased in mddcc is limited, our findings support that intronic DNA methylation is required for the proper expression of some Intron teM genes by promoting the production of its full length transcript" is really bad. Change into "Our findings suggest that intronic DNA methylation is required for the proper expression of some Intron teM genes by promoting the production of its full-length transcript, although the number of teM genes, for which 3' transcripts are decreased in mddcc, is limited."

Response: We thank the reviewer for this suggestion; we have changed the sentence as suggested (Lines 252-255).

Reviewer #5 (Remarks to the Author):

I have carefully reviewed the revised manuscript and found that authors were able to address many of the reviewers' comments, through rewording and clarifying confusing or misleading statements. One critical issue remains regarding the use of 5-week-old mddcc plant for RNA-seq comparison with 2-week-old wt, met1, ddcc plants. The suggestion by reviewer #3 for better normalization with spike-in mRNA controls was not properly addressed. The purpose of spike-in control relative to same # of input cells was to address the assumption that each cell generates same amount of total RNAs. The authors should review this important paper on why spike-in is critical for global gene expression analysis (Lovén et al (2012) Cell 151 (3), 476-482). The authors used qRT-PCR of selected 25 up-

and down-regulated genes plus the use of DESeq2 for normalization would be satisfactory if each cell from different genotype does produce same amount of RNAs, which may well be true but is an unproven assumption.

Response: Indeed, both qRT-PCR and DESeq2 are used for normalization under some unproven assumptions: qRT-PCR assumes that each cell from different genotypes produces the same amount of internal control RNAs; DESeq2 assumes that most genes show a similar expression level among different genotypes. To our knowledge, DESeq2 (Love et al., 2014, Genome Biology) has been most frequently used for DEG analysis in the RNA-seq community (19,948 citations to date). qRT-PCR is a commonly used method to independently validate the results from RNA-seq. Importantly, our results from RNA-seq and qRT-PCR are consistent, suggesting that our identified DEGs in *ddcc*, *met1-9*, and *mddcc* are reliable.

The spike-in control is undoubtedly a robust method for normalization. However, it works under the additional prerequisite that the cell numbers used for RNA-seq must be the same among different genotypes (Lovén et al., Cell 2012), which is extremely challenging for plant tissues, and consequently spike-in controls have rarely been used for RNA-seq analysis in plants.

There is another important issue that needs to be addressed, i.e., the use of different age materials for *wt/met1/ddcc* (2wo) and *mddcc* (5wo). Since several major conclusions were drawn from transcriptome results, simply includes a disclaimer in the discussion that “this difference in sampling time might have unintended, secondary effects, which must be considered when interpreting the results” (Lines 435-436) is unacceptable. The authors have to do more to demonstrate *mddcc* DEGs are induced in WT at older age so that their expression change can truly be ascribed to the loss of DNA methylation. I understand the difficulty of such experiments. I think one possible way is to compare transcriptomes of *wt* plants between 2W and 5W of age, even just the use of rosette leaves would be a good start. The authors should also perform 2W-2W (*wt-mut*) and 2W-5W (*mut-mut*) comparisons to filter out genes whose expression changes over development.

Response: We thank the reviewer for this suggestion. Following the reviewer’s advice, we have included RNA-seq data of 5-week-old WT plants for comparison. The total RNA of 5-week-old WT plants was extracted from the aerial part after removing the inflorescence. We were not able to generate RNA-seq data of 2-week-old *mddcc* plants because collecting enough 2-week-old *mddcc* plant tissue is very challenging since they are extremely small, hence the available tissue is very limited and they must be genotyped beforehand. To address the reviewer’s concern, we examined the relative expression of previously identified DEGs (5-week-old *mddcc* relative to 2-week-old WT) in 5-week-old *mddcc* compared to 5-week-old WT plants. We found that more than half of these DEGs are similarly up/down-regulated in 5-week-old *mddcc* compared to 5-week-old WT plants (Supplementary Fig. 19). Most of those DEGs that are not similarly up/down-regulated in 5-week-old *mddcc* compared to 5-week-old WT plants display the corresponding changes in 2-week-old *met1* plants relative to 2-week-old wild-type plants (Supplementary Fig. 19), suggesting that the expression of these genes can be regulated by DNA methylation

rather than purely developmental stage-dependent. In parallel, we compared the expression levels of selected previously described DEGs potentially involved in specific phenotypes of the *mdgcc* mutant, finding that the majority of very highly up-regulated DEGs are also up-regulated in 5-week-old *mdgcc* relative to 5-week-old WT plants, despite the differences in developmental stage (Supplementary Fig. 20). Taken together, these results indicate that the transcript differences identified between the *mdgcc* mutant and WT are, at least to a large extent, not artifactual due to the age difference. Admittedly, we cannot exclude that the sampling time might affect the identification of DEGs in *mdgcc*, as specifically indicated in the text as a cautionary note.

We have modified this paragraph “It should be noted that the age of the plants used for the transcriptomic analysis is genotype-dependent: while two-week-old plants were used for the WT, *met1*, and *ddcc*, five-week-old plants were used for the quintuple *mdgcc* mutant. The justification for this experimental design lies on the extremely slow pace of growth and development exhibited by the latter, which renders five-week-old *mdgcc* plants similar to two-week-old plants of the other genotypes analyzed. Nonetheless, this difference in sampling time might have unintended, secondary effects, which must be considered when interpreting the results.” to “It should be noted that the age of the plants used for the transcriptomic analysis is genotype-dependent: while two-week-old plants were used for the WT, *met1*, and *ddcc*, five-week-old plants were used for the quintuple *mdgcc* mutant. The justification for this experimental design lies on the extremely slow pace of growth and development exhibited by the latter, which renders five-week-old *mdgcc* plants more similar to two-week-old plants of the other genotypes analyzed. To exclude the possibility that DEGs identified in *mdgcc* in this study are mainly caused by the difference in sampling time, we compared the expression levels of these DEGs between 5-week-old *mdgcc* and 5-week-old WT plants. We found that more than half of these DEGs are similarly up/down-regulated in 5-week-old *mdgcc* compared to 5-week-old WT plants (Supplementary Fig. 19). Most of those DEGs that are not similarly up/down-regulated in 5-week-old *mdgcc* compared to 5-week-old WT plants display the corresponding changes in 2-week-old *met1* plants relative to 2-week-old wild-type plants (Supplementary Fig. 19), suggesting that the expression of these genes can be regulated by DNA methylation rather than purely developmental stage-dependent. In parallel, we compared the expression levels of selected previously described DEGs potentially involved in specific phenotypes of the *mdgcc* mutant, finding that the majority of very highly up-regulated DEGs are also up-regulated in 5-week-old *mdgcc* relative to 5-week-old WT plants, despite the differences in developmental stage (Supplementary Fig. 20). Taken together, these results indicate that the transcript differences identified between the *mdgcc* mutant and WT are, at least to a large extent, not artifactual due to the age difference. Nevertheless, an effect of the sampling time on the transcript differences observed cannot be excluded” (Lines 448-464).

Supplementary Figure 19. Expression profile of DEGs in 5-week-old *mddcc* relative to 2-week-old WT among the indicated genotypes.

Supplementary Figure 20. Comparing the expression levels of selected DEGs (5-week-old *mdcc* relative to 2-week-old WT) potentially involved in the specific phenotypes of *mdcc* among the indicated genotypes.

The authors should do a better job stating their results around the novelty of the genetic material rather than the novelty of their results to prevent criticism of over exaggeration of the significance of their findings. I have pointed out some wording suggestions below, along with specific comments of the manuscript that I hope can helpful during their revision.

Specific comments:

mdcc vs *ddcc*, were the *ddcc* plants derived from *m/+ddcc* sergeants? This was not clear in the main text and M&M section.

Response: The *ddcc* mutant used in this manuscript was generated by genetic crossing (*cmt2* x *ddc*) and subsequent PCR-based genotyping in F2 populations. This is now specifically mentioned in the Methods section (Lines 525-528).

Lines 33-35 “however, the extent of involvement of this epigenetic mark in genome-wide gene regulation and plant developmental control is unclear.” The word “unclear” is not appropriate. This gives a false sense of not much is known. This sentence should be rephrased to reflect the true nature of this manuscript, i.e., the effect of genome-wide gene regulation and plant development in a methylation-free environment.

Response: Following the reviewer’s suggestion, we have changed the sentence to “however, our understanding of the extent of involvement of this epigenetic mark in genome-wide gene regulation and plant developmental control is incomplete” in lines 33-35.

Line 38-39, “.CG methylation and non-CG methylation alone or jointly are required for a plethora...” the work alone should be removed.

Response: Following the reviewer’s suggestion, the word “alone” has been removed (Line 41).

Line 117-120, suggest changing to something like “Our findings extend prior knowledges on the importance of DNA methylation for global gene expression and development.....”

Response: We have changed this sentence following the suggestion of the reviewer (Lines 119-120).

Line 164 impact of CG methylation on gene transcription directly and indirectly....

Response: We have revised the sentence as suggested (Line 167).

Line 169, the qPCR validation for up-regulate

Response: We changed the “RT-qPCR” to “qPCR” in line 173.

M&M Defining DNA methylation-associated genes. The description was confusing regarding whether promoter methylation (pM genes) was calculated using 2kb or just 500-bp. (lines 641-642)

Response: We apologize for the confusion. The 2 kb-sequence upstream of genes was defined as the promoter. The total number of C and C + T at all cytosine positions within each 500-bp bin of each gene promoter were computed, and the region with maximum DNA methylation level was selected for subsequent analysis. We have modified “gene promoter” to “gene promoter (2 kb)” (Line 675).

Fig 2e, bottom panel, these are the UM genes but how come they have CG methylation in WT and in *ddcc*? Did I miss anything? This is also seen in Fig S3b,c bottom panels.

Response: The classification of genes was based on their methylation level in the WT background (Bewick et al., PNAS 2016). The methylation level of genes must reach the threshold (see the method, q value < 0.01) to be defined as methylated genes in our analysis. It is expected that genes with very low/trace levels of DNA methylation do not pass this statistical test. Accordingly, those genes were defined as UM genes.

Lines 274, AT2TE42810...., shouldn't this be AT5TE65370 instead? As AT2TE42810 expression shows no difference between *met1* and *mddcc*.

Response: Even though the overall levels of expression of AT2TE42810 are similar between *met1* and *mddcc*, we found that a partial transcript of AT2TE42810 is increased in *mddcc* relative to *met1* (Fig. 3j, see the wireframe). Thus, we stated that “Interestingly, we found that only a partial and antisense transcript of AT2TE42810 is increased in *mddcc* relative to *met1-9*, implying that the activation of this part of the locus may be more important than the other part for AT2TE42810 transposition (Fig. 3j)”.

Line 280-281, “Our findings demonstrate that CG and non-CG methylation act redundantly to prevent TE transposition”. Such statement sounded as if this is a new discovery but in fact this is at best an independent confirmation for many prior results. This should be restated accordingly.

Response: Following the reviewer's suggestion, we have modified this sentence to “Our findings confirm prior results indicating that CG and non-CG methylation act redundantly to prevent TE transposition” (Line 288).

Lines 331-333, what tissue was used for flow cytometry study? The M&M said 11-d-old cotyledon, but the main text says mesophyll cell? Please clarify. If the authors can compare 11-day old nuclei, why can't they compare transcriptome of the same stage of these mutants?

Response: DNA content was determined by flow cytometry using cotyledons. We changed “mesophyll cell” to “nuclei isolated from cotyledons from 11-day-old seedlings” in line 339. This experiment only requires a very small amount of cotyledon tissues; however, the required amounts are much higher for RNA-seq (see also above response).

Lines 371-, regarding the 3 other MET1 orthologs, their expressions are very low during development but are somewhat induced during reproduction, particularly during endosperm development (see Hsieh et al, PNAS, 2011, PMID: 21257907).

Response: We thank the reviewer for bringing this to our attention. We have now modified the corresponding sentence to “Considering that these homologous genes are expressed in seedlings (Winter et al., PLoS ONE 2007) and that DNA methylation is completely erased in *mdgcc*, it seems likely that the encoded proteins do not play a prevalent role in maintaining DNA methylation, although we cannot exclude that they may maintain DNA methylation in endosperm since their expression levels are up-regulated in endosperm relative to other tissues (Hsieh et al., PNAS 2010)” (Lines 381-387).

Lines 394-397, the authors should discuss recent papers on DNA methylation reader complex) PMID: 30523112, 34185870, 26400170, 30589221)

Response: Following the reviewer's suggestion, we have added the following sentences to the Discussion section (lines 409-410): “Recently discovered DNA methylation reader complexes which promote gene expression support this assumption (Li et al., Nucleic Acids Res 2015; Harris et al., Science 2018; Zhao et al., J Integr Plant Biol 2019; Xiao et al., J Integr Plant Biol 2019; Miao et al., Plant Cell Physiol 2021)”.

Line 400-401, “... alternatively, is regulated by distant DNA methylation through chromosomal interaction” This does not make much sense to me. Please elaborate.

Response: We assume a situation where repressors/activators bind to methylated DNA and regulate the expression of the distant unmethylated gene through the DNA loops formed by chromosomal interactions.

Reviewers' Comments:

Reviewer #1:

Remarks to the Author:

The authors have considered my points from the previous re-review, and I have no further concern. However, in the meantime, a publication appeared that addresses a similar question as here. Liang et al., *New Phytologist* 2021, "Deciphering the synergistic and redundant roles of CG and non-CG DNA methylation in plant development and TE silencing" also describe attempts to generate a quintuple mutants lacking the five DNA methyltransferases. In their case, the quintuple mutant is described as embryo-lethal. This may be due to the nature of the respective mutant alleles, but it needs at least to be discussed and the other publication cited.

Reviewer #5:

Remarks to the Author:

In this revision, minor comments were addressed, and the overall presentation is improved. However, I am afraid the including of Fig S19 and S20 are not sufficient to address earlier concerns comparing samples from different developmental stages. The main purpose of including 2W-5W (wt) comparison is to filter out genes with transcriptional change in response to development so other analyses can be done again with a set of high confident DEGs. The authors present a heatmap view of additional genotypes and reasoned that a subset previously identified DEGs (5-week-old mddcc relative to 2-week-old WT), even though they are also changed in 5W wt vs 2W wt, they are also responsive (expression change) in 2W met1 sample. Although I understand the argument, the data presented are descriptive in nature. I have also come to the realization that this is a hard problem to tackle. This intrinsic nature of the samples makes it impossible to decouple methylation effect from the developmental effect. Unfortunately, this makes most of the other analyses throughout the manuscript descriptive and qualitative, rather than an unequivocal demonstration of a complete loss of DNA methylation and their effects on gene transcription. Although the authors have repetitively stressed the difficulty of obtaining enough materials from mddcc plants, to me it might be less of a problem to compare samples with the same developmental stage. Plus, techniques exist that can use very low input materials for transcriptome analysis that should be considered along with proper spike-in controls to make this a more robust comparison and analysis. I am also not entirely sure whether the RNA-seq analysis done by the authors (not sufficiently described in the M&M) permit multiple sample comparison using heatmaps. A newly published paper in *New Phytologist* (Liang et al, 2021) reported the same mddcc plants but showed quintuple mutants are embryo lethal. The authors should at least discuss the apparent discrepancy in their future revision.

Point-by-Point response to reviewers

We would like to thank the reviewers for their careful review of our manuscript. We sincerely believe that we have improved our manuscript owing to their comments and suggestions. A point-by-point response to the reviewers' comments is provided below.

Reviewer #1 (Remarks to the Author):

The authors have considered my points from the previous re-review, and I have no further concern.

However, in the meantime, a publication appeared that addresses a similar question as here. Liang et al., *New Phytologist* 2021, "Deciphering the synergistic and redundant roles of CG and non-CG DNA methylation in plant development and TE silencing" also describe attempts to generate a quintuple mutants lacking the five DNA methyltransferases. In their case, the quintuple mutant is described as embryo-lethal. This may be due to the nature of the respective mutant alleles, but it needs at least to be discussed and the other publication cited.

Response: We thank the reviewer for this suggestion; we have added the following sentences to the Discussion section (lines 391-397): "While our manuscript was under review, Liang et al. reported that an *mddcc* mutant is embryonically lethal (Liang et al., 2022, *New phytol*). It should be pointed out that their failure to obtain a viable *mddcc* mutant likely results from the fact that their selected *ddcc* alleles, as well as the process employed for the generation of the quintuple mutant (Liang et al., 2022, *New phytol*), are different from the ones used in this work."

Reviewer #5 (Remarks to the Author):

In this revision, minor comments were addressed, and the overall presentation is improved. However, I am afraid the including of Fig S19 and S20 are not sufficient to address earlier concerns comparing samples from different developmental stages. The main purpose of including 2W-5W (wt) comparison is to filter out genes with transcriptional change in response to development so other analyses can be done again with a set of high confident DEGs. The authors present a heatmap view of additional genotypes and reasoned that a subset previously identified DEGs (5-week-old *mddcc* relative to 2-week-old WT), even though they are also changed in 5W wt vs 2W wt, they are also responsive (expression change) in 2W *met1* sample. Although I understand the argument, the data presented are descriptive in nature. I have also come to the realization that this is a hard problem to tackle. This intrinsic nature of the samples makes it impossible to decouple methylation effect from the developmental effect. Unfortunately, this makes most of the other analyses throughout the manuscript descriptive and qualitative, rather than an unequivocal demonstration of a complete loss of DNA methylation and their effects on gene transcription. Although the authors have repetitively stressed the difficulty of obtaining enough materials from *mddcc* plants, to me it might be less of a problem to compare samples with the same developmental stage. Plus, techniques exist that can use very low input materials for transcriptome analysis that should be considered along with proper spike-in controls to make this a more robust comparison and analysis. I am also

not entirely sure whether the RNA-seq analysis done by the authors (not sufficiently described in the M&M) permit multiple sample comparison using heatmaps. A newly published paper in *New Phytologist* (Liang et al, 2021) reported the same *mdcc* plants but showed quintuple mutants are embryo lethal. The authors should at least discuss the apparent discrepancy in their future revision.

Response: We thank the reviewer for this comment and suggestion. Indeed, the intrinsic nature of *mdcc* mutant (extreme developmental defects) makes it difficult to distinguish between DNA methylation and developmental effects on gene expression. We need to point out that the developmental stage of *mdcc* is extremely delayed relative to 2-week-old WT. Thus, we believe providing the RNA-seq data of 2-week-old *mdcc* would not help reduce the potential developmental effects on gene expression. This was our rationale to choose the 5-week-old *mdcc*, whose developmental stage is closer to other genotypes at 2 weeks post-germination, to generate the RNA-seq data (as Reviewer 1 previously stated). Nevertheless, we have added the following cautionary note to the Discussion section (lines 461-463): “Moreover, the difference in the development of *mdcc* mutant compared to other genotypes could have unintended, secondary effects, which must be considered when interpreting the results”.

The clustering in heatmaps is based on all of the displayed samples.

Following the reviewer’s advice, we have added these sentences to the manuscript: “While our manuscript was under review, Liang et al. reported that an *mdcc* mutant is embryonically lethal (Liang et al., 2022, *New phytol*). It should be pointed out that their failure to obtain a viable *mdcc* mutant likely results from the fact that their selected *ddcc* alleles, as well as the process employed for the generation of the quintuple mutant (Liang et al., 2022, *New phytol*), are different from the ones used in this work.” (lines 391-397).